# Structure dynamics of HIV-1 Env trimers on native virions engaged with living T cells

Irene Carlon-Andres [1,2,3✉], Tomas Malinauskas [3] & Sergi Padilla-Parra [1,2,3✉]

The HIV-1 envelope glycoprotein (Env) mediates viral entry into the host cell. Although the highly dynamic nature of Env intramolecular conformations has been shown with single molecule spectroscopy in vitro, the bona fide Env intra- and intermolecular mechanics when engaged with live T cells remains unknown. We used two photon fast fluorescence lifetime imaging detection of single-molecule Förster Resonance Energy Transfer occurring between fluorescent labels on HIV-1 Env on native virions. Our observations reveal Env dynamics at two levels: transitions between different intramolecular conformations and intermolecular interactions between Env within the viral membrane. Furthermore, we show that three broad neutralizing anti-Env antibodies directed to different epitopes restrict Env intramolecular dynamics and interactions between adjacent Env molecules when engaged with living T cells. Importantly, our results show that Env-Env interactions depend on efficient virus maturation, and that is disrupted upon binding of Env to CD4 or by neutralizing antibodies. Thus, this study illuminates how different intramolecular conformations and distribution of Env molecules mediate HIV-1 Env–T cell interactions in real time and therefore might control immune evasion.

[1] Department of Infectious Diseases, King's College London, Faculty of Life Sciences & Medicine, London, United Kingdom. [2] Randall Division of Cell and Molecular Biophysics, King's College London, London, United Kingdom. [3] Division of Structural Biology, Wellcome Centre for Human Genetics, University of Oxford, Oxford, United Kingdom. ✉email: irene.carlon-andres@kcl.ac.uk; sergio.padilla_parra@kcl.ac.uk

HIV-1 entry into the cell requires fusion of the viral lipidic envelope with the host plasma membrane. This process is mediated by the HIV-1 Env glycoprotein, which is the sole protein embedded in the viral lipid envelope and is found in low density per virion (7–14 spikes per particle)[1–3]. It is encoded in the viral gene *env* as the precursor transmembrane protein gp160. Processing of gp160 by the cellular protease furin and subsequent trafficking from the ER to the plasma membrane results in a glycosylated homotrimer of non-covalently bound gp120-gp41 heterodimers[4]. The extracellular subunit gp120 consists of five conserved regions (C1-C5) and five variable regions (V1-V5) and is responsible of receptor and co-receptor binding: the host receptor CD4 binds to conserved regions flanking V4 whereas the coreceptor CCR5 binds to V3[5]. The subunit gp41 contains a transmembrane domain and a cytoplasmic tail (CT), which in turn, binds the matrix domain of the viral precursor Gag during virion assembly. Virions are released from the host cell as immature viral particles. Further cleavage of the HIV-1 Gag and Pol precursor proteins by the viral protease is required for the formation of fully mature infectious virions. Indeed, interactions between CT of gp41 and unprocessed Gag have been shown to impair Env fusion activity[6,7]. Consistently, Env clustering on the surface of virions has been only observed in mature virions and cluster formation has been shown to depend on the CT of gp41[8].

The interaction between HIV-1 Env and host receptor CD4 triggers a series of conformational changes allowing the co-receptor, either CCR5 or CXCR4, to bind the prefusion complex via a common three-step mechanism, in which one or two Env protomers engage in the prefusion complex[9,10]. The gp41 protein is then inserted in the host plasma membrane forming a pre-harpin conformation that will eventually lead to the formation of the fusion pore[3]. Fusion of the viral lipid envelope with the plasma membrane ultimately enable the viral core to access the cellular replication machinery.

The structure and intramolecular dynamics of the HIV-1 Env have been extensively studied during the past few years. These studies have benefited from structural approaches such as x-ray crystallography[11,12] and cryo-electron microscopy[4,13–17], and single-molecule FRET techniques[18–21]. High-resolution structures have been elucidated using a soluble and stabilized version of the HIV-1 Env (SOSIP), which provided information about the pre-fusion and CD4 bound intramolecular conformations of Env[5,14–17]. Molecular characterization of broad neutralizing antibodies (bNAbs) targeting different epitopes of Env has been also a valuable tool to resolve the stabilized pre-fusion closed state, receptor-bound open state and to identify different intermediate states of the Env trimer[13,14,22–25]. Neutralizing antibodies can be classified based on their epitope: CD4-binding site (i.e. b12 antibody, which stabilize the trimer in an intermediate open conformation), membrane-proximal external region (MPER) (i.e. 10E8, which stabilizes an open Env conformation) or antibodies recognizing the apex of Env (i.e. PGT145, which recognizes a closed Env conformation) among others[26].

Munro et al.,[20] pioneered in the study of in vitro intramolecular structural dynamics of fluorescently labelled HIV Env trimers in native virions utilizing single-molecule Förster Resonance Energy Transfer (FRET). This technique revealed the dynamic fluctuations of the distance between fluorescently labelled residues within the Env trimer in a millisecond-second timescale. These studies performed in the native, virion-associated Env of different HIV-1 strains helped to describe three different intramolecular conformational states of Env: first, Env adopts a closed conformation (named State 1) right before CD4 asymmetric interaction; second, after CD4 engagement, Env adopts an intermediate state (State 2) followed by a last open conformation

for the coreceptor engagement (State 3) that exposes otherwise hidden epitopes, increasing susceptibility for antibody recognition[18–20].

It is currently unclear how different intramolecular Env conformations can be reconciled with Env diffusion[2] and intermolecular dynamics[8] during cluster formation and dissociation in mature HIV-1 viruses and its relation with the prefusion reaction on the surface of the host. Moreover, it is still not clear whether these three intramolecular states described in vitro recapitulate the bona fide dynamics of HIV-1 Env when engaged with live T cells, in the presence or absence of broadly neutralizing antibodies.

In this study, we were able to detect intramolecular conformational states of Env, also described before[20], and to determine how they relate to Env cluster distribution within the viral membrane during the first steps of the HIV-1 prefusion reaction with living T cells. We also describe the role of Env-Env dissociation when exposed to broadly neutralizing antibodies.

## Results

**Characterization of HIV-1 Env structural dynamics by two photon FRET-FLIM.** Aiming to ascertain both intramolecular and intermolecular Env dynamics with a multiparameter FRET and fluorescence lifetime microscopy (FLIM) approach, we produced HIV-1 virions labelled with super folder GFP (GFP$_{OPT}$) in the V4 loop of gp120 HXB2 Env glycoprotein (Fig. 1a–c)[27]. HIV-1 virions pseudo-typed with HXB2 V4-GFP$_{OPT}$ Env were exposed to monoclonal nanobodies against GFP, in turn, labelled with Atto 488 (NbA488) and Atto 594 (NbA594), that constitute the donor and acceptor dipoles of the FRET pair, respectively (Fig. 1c). This particular labelling strategy allows FRET to occur between donor and acceptor dipoles located in a single Env molecule (intramolecular interaction) and between adjacent Env molecules (intermolecular interaction), when fluorophores are in close enough proximity and in a proper orientation (Fig. 2a). To be certain of only considering bona fide HIV-1 virions and being able to determine their maturation state for subsequent FRET-FLIM analysis, pseudo-virions were produced harbouring the Gag polyprotein precursor fused to GFP (Fig. 1a, b). Virion labelling efficiency was determined by exposing HIV-1$_{Gag-GFP\ HXB2\ V4-GFPOPT}$ virions to NbA594 and quantifying the percentage of double-positive (GFP + NbA594 + ) particles, which was 32.7% of the total GFP + particles ($n = 1155$) (Fig. 1d). Importantly, it was previously shown that labelling the V4 loop of gp120 with GFP$_{OPT}$ does not significantly interfere with HIV-1 fusion activity[28]. We also evaluated the ability of HIV-1$_{Gag-GFP\ HXB2\ V4-GFPOPT}$ viruses to infect TZM-bl reporter cells compared to HIV-1$_{Gag-GFP}$ bearing the wildtype HXB2 envelope protein using an X-gal infectivity assay (Fig. 1e–g). TZM-bl cells were exposed to equivalent amounts of viral particles from a mature-enriched (viruses produced in absence of the HIV-1 protease inhibitor Saquinavir, SQV) or immature viral sample (produced in presence of SQV) at different dilutions, as indicated in Fig. 1f. Labelling of HXB2 in the V4 loop of gp120 did not statistically significantly impair viral infectivity compared to viruses pseudo-typed with WT HXB2 envelope (Fig. 1g). In contrast, inhibition of the HIV-1 protease by SQV treatment dramatically disrupted viral infectivity regardless of the viral labelling strategy (Fig. 1e, f).

We also assessed the maturation efficiency of the viral sample by tracking the release of the internal GFP after exposure to a 0.01% concentration of saponin. Only mature HIV-1 particles that undergo proteolytic processing of Gag are able to release the GFP content after permeabilization of the viral membrane. Double positive (GFP + NbA594 + ) mature virions showed a drop in GFP fluorescence upon saponin treatment and a stable Atto594

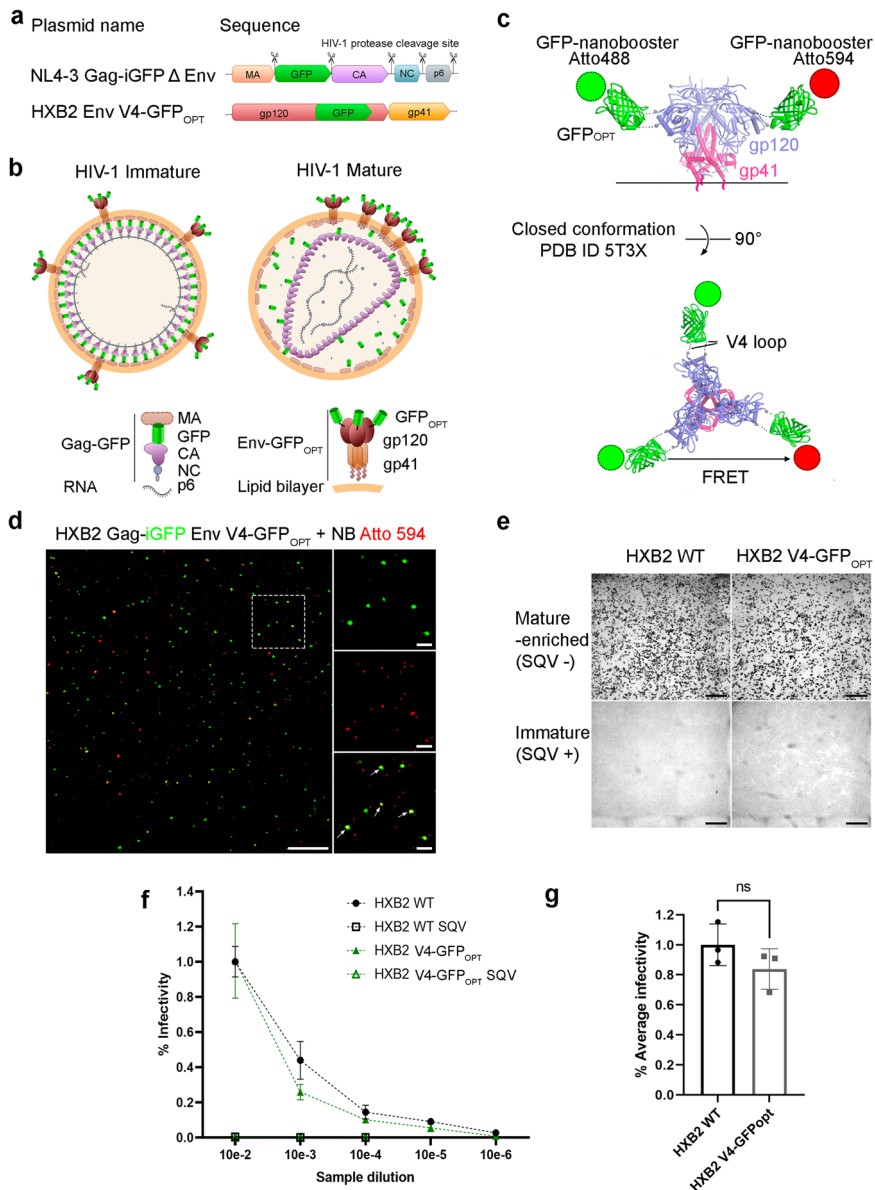

**Fig. 1 HIV-1 HXB2 labelling strategy and functionality of labelled viruses. a** Diagram representing the DNA sequence of the plasmids used to produce HXB2 pseudotyped virions. MA: matrix. GFP: green fluorescent protein. CA: capsid. NC: nucleocapsid. Protease cleavage sites are indicated by the scissors symbol. **b** Schematic representation of HIV-1 immature and mature particles used in this assay. Immature particles possess the unprocessed Gag polyprotein fused to GFP. The viral protease cleaves Gag to mediate assembly of the mature HIV-1 virion. Few copies of HXB2 V4-GFP$_{OPT}$ embedded in the viral membrane allow analysis of Env conformations by FRET-FLIM and single-molecule approaches. **c** The HIV-1 Env structure representing closed Env (PBD ID 5T3X) conformation was modified to illustrate the labelling of the V4 loop with GFP$_{OPT}$ and NbA488 and NbA594. **d** Micrograph showing fluorescent HIV-1 pseudoviruses expressing Gag-GFP and HXB2 V4-GFP$_{OPT}$ (green) labelled with Atto 594 (red). Scale bar 10 µm. Magnification of the region delimited with a dashed contour is shown in the right column. Scale bar 2 µm. Arrows point to colocalisation events (GFP+ Atto 594+, yellow particles) which represent efficient labelling of virions. **e** Micrograph from transmission microscopy of TZM-bl cells infected with HIV-1 Gag-GFP viruses pseudotyped with WT HXB2 or labelled HXB2 produced in the absence (mature-enriched) or presence (immature) of the HIV-1 protease inhibitor SQV. Infected cells show a characteristic dark contrast as a result of X-gal hydrolysis upon β-galactosidase expression occurring in infected TZM-bl cells. Scale bar 0.5 mm. **f** Quantification of the infection efficiency (% of infected cells) in TZM-bl cells infected with equivalent amounts of HIV-1 pseudotyped viruses at different dilutions, as indicated. Dots represent the mean from three independent experiments and error bars the SD. **g** Quantification of the relative infection efficiency of the mature-enriched HXB2 V4-GFP$_{OPT}$ sample compared to the WT HXB2 pseudo-typed HIV-1 viruses. Bars represent the mean of $n = 3$ independent experiments, dots the individual values and error bars the SD. Unpaired two-tailed $T$-test (ns: $p = 0.2213$).

fluorescence signal over time (Supplementary Fig. 1a, top panel), showing both, a negligible contribution of photons from the V4-GFP$_{OPT}$ Env and high stability of the Atto594 fluorophore, making this labelling suitable for our FRET-FLIM experiments. In contrast, immature virions (+SQV) did not show a drop in GFP fluorescence over time but an increase in Atto594 fluorescence intensity,

suggesting increased accessibility of the NbA594 to the unprocessed Gag-GFP after viral membrane permeabilization (Supplementary Fig. 1a, bottom panel). We observed that $34.3 \pm 8.2\%$ of virions produced in absence of the HIV-1 protease inhibitor SQV were able to release the internal GFP against 0%, when virions were produced in presence of SQV (Supplementary Fig. 1b).

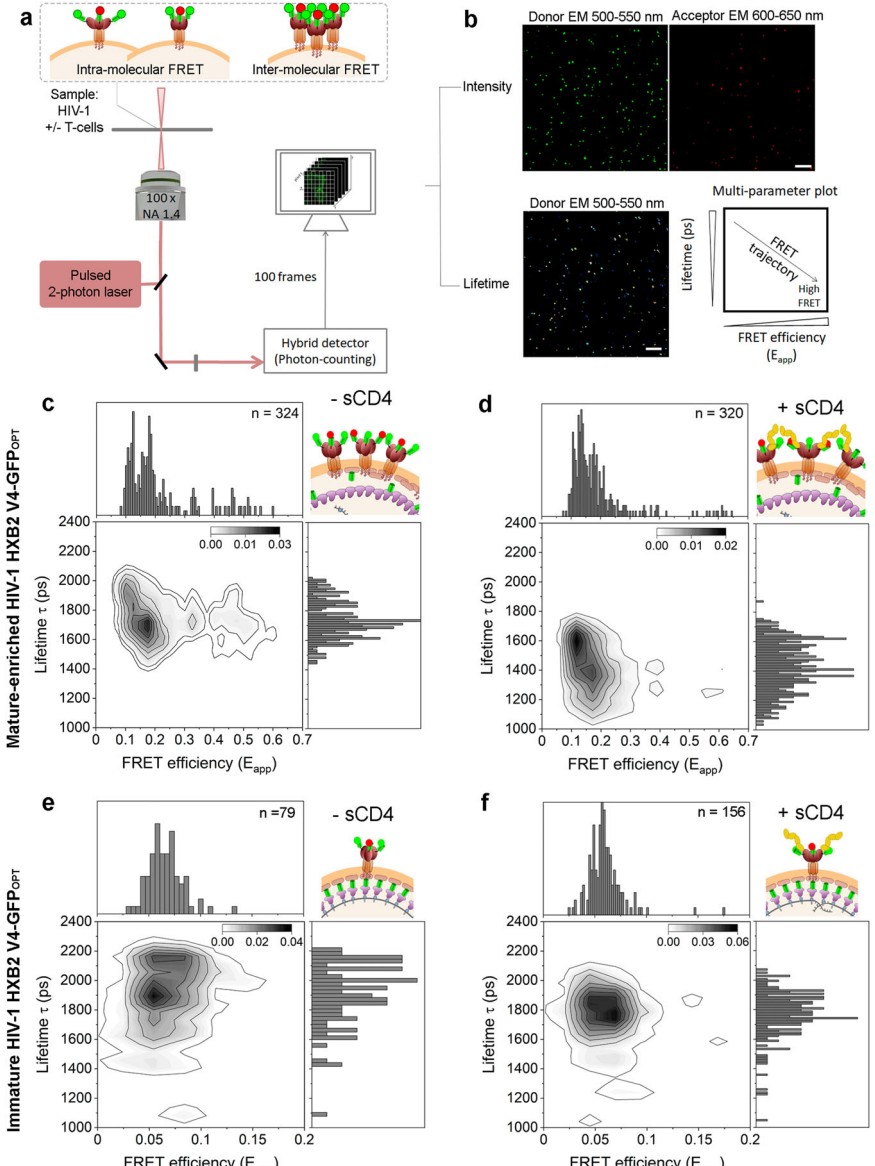

**Fig. 2 Two Photon FRET-FLIM detects HIV-1 Env conformations. a** Schematic representation of the FRET-FLIM system setup. Intramolecular FRET between Env trimer labels is resolved in sparse Env molecules whereas inter-molecular FRET is observed upon Env-Env association. Samples of HIV-1 virions in the presence or absence of T cells were imaged using a Leica DIVE two-photon system equipped with a pulsed two-photon laser tunned at 950 nm, a 100× magnification objective and photon-counting technology. The simultaneous intensity and lifetime measurements were obtained in 100 frames acquisitions (see material and methods for further details). **b** Micrographs showing images corresponding to intensity-based measurements obtained in two independent channels: donor emission (500-550 nm) and acceptor emission (600–650 nm) or to lifetime measurements of the donor. Scale bar 10 μm. Data from donor lifetime (ps) and calculated apparent FRET efficiency from intensity-based measurements is represented in kernel multi-parameter plots. In presence of both, donor and acceptor, FRET values experience a shift towards positive values concomitant with a drop in lifetime. (**c**-**f**) Two-dimensional (2D) kernel probability graphs showing FRET (FRET efficiency, $E_{app}$) vs FLIM (Lifetime, in ps) data. High-density regions are depicted in darker grey color. **c**, **d** Plots corresponding to HIV-1 mature HXB2 V4-GFP$_{OPT}$ labelled with NbA488 and NbA594 nanobodies incubated without (**c**) or with (**d**) sCD4. **e**, **f** Plots corresponding to HIV-1 immature HXB2 V4-GFP$_{OPT}$ labelled with NbA488 and NbA594 nanobodies incubated without (**e**) or with (**f**) 10 μg/mL concentration of soluble CD4 (sCD4$_{D1-D4}$).

**Intermolecular Env dynamics are maturation-sensitive and modulated by sCD4.** Labelled HIV-1$_{Gag-GFP}$ HXB2 V4-GFPOPT virions were imaged using two-photon rapid FLIM[29,30] (Fig. 2a). Viral samples were imaged during 5 min with a 3 s interval between frames (100 frames per acquisition). Both fluorescence lifetime and intensity-based measurements were simultaneously acquired. Average donor lifetime and apparent FRET efficiency ($E_{app}$) were calculated per acquisition and results from single viral particle analyses were plotted into multi-parameter two-dimensional Kernel graphs[31] (Fig. 2b). FRET is a dipole-dipole coupling

process where the energy is transferred from the excited donor fluorophore to the acceptor fluorophore when the distance and orientation of both dipoles are the right ones (typically within 10 nm and a random orientation). The excitation of the donor fluorophore induces a sensitized emission from the acceptor concomitantly quenching the fluorescence of the donor. This process, in the absence of acceptor would not occur. Therefore the average intensity-based $E_{app}$ reveals proximity between donor and acceptor. In turn, the fluorescence lifetime of the donor in the presence of an acceptor is sensitive to changes in FRET but also in

the local microenvironment conditions. Hence, the multi-dimensional analysis of correlated changes of FRET and FLIM allows to efficiently detect heterogeneities in the dynamics of a given population even in low-photon conditions, where few labelled proteins are available[32,33].

To define the heterogenous Env conformational landscape by FRET-FLIM, HIV-1$_{Gag-GFP\ HXB2\ V4-GFPOPT}$ particles were exposed to equimolar concentrations of both, NbA488 (donor) and NbA594 (acceptor), which specifically target external GFP$_{OPT}$ in the viral Env. If FRET would occur between Env V4 labels (NbA488 and NbA594), the fluorescence lifetime of NbA488 in the presence of NbA594 would be shortened or quenched and the apparent FRET efficiency would be increased[34,35] (Fig. 2b). As expected, the addition of both, donor and acceptor fluorophores (NbA488 and NbA594, respectively) induced a shift towards positive $E_{app}$ values relative to the spectral profile observed in HIV-1$_{Gag-GFP\ HXB2\ V4-GFPOPT}$ particles exposed to the donor fluorophore alone (Nb488) (Supplementary Fig. 2): $E_{app} > 0.1$, in case of the mature-enriched HIV-1$_{Gag-GFP\ HXB2\ V4-GFPOPT}$ sample (Fig. 2c, d) and $E_{app} > 0.06$ in immature particles (Fig. 2e, f). Importantly, this shift towards positive $E_{app}$ values was concomitant with a decrease in lifetime values.

FRET-FLIM analysis of the mature-enriched HIV-1$_{Gag-GFP\ HXB2\ V4-GFPOPT}$ sample in presence of donor and acceptor fluorophores yielded three main FRET regimes (Fig. 2c): i) low or no FRET ($E_{app} < 0.12$) and higher lifetimes (~1900 ps) ii) a more dense population showing intermediate FRET efficiency ($0.12 < E_{app} < 0.23$) and moderately decreased lifetimes (~1750 ps), and iii) high apparent FRET efficiency ($E_{app} > 0.23$) and decreased lifetimes (~1700 ps).

Seeking to relate the observed FRET-FLIM profile with intramolecular conformations of HIV-1 Env in our functional virions, we exposed a viral sample of mature-enriched HIV-1$_{Gag-GFP\ HXB2\ V4-GFPOPT}$ particles, to saturating concentrations (10 μg/mL) of soluble CD4 (sCD4$_{D1-D4}$) (Fig. 2d). We could readily stabilize a low or no-FRET situation ($E_{app} < 0.12$) that we could assign to an Env open conformation as seen by others[36], whereas the intermediate and the high FRET regime populations were clearly reduced. This result suggests that intermediate ($0.12 < E_{app} < 0.23$) and high FRET regimes ($E_{app} > 0.23$) could relate to an intramolecular closed Env conformation from sparse Env molecules or intermolecular Env-Env interactions, in which the conditions for FRET to occur would be more favourable.

It has been previously shown that in immature HIV-1 viruses, Env diffuses twice as slow ($D = 0.001\ \mu m^2/sec$) as compared to mature HIV-1 particles ($D = 0.002\ \mu m^2/sec$)[2]. Moreover, in immature HIV-1 virions, Env molecules are sparsely distributed in the viral envelope[8]. Based on these observations and to accurately define a FRET threshold for intramolecular interactions, we produced immature HIV-1 virions[37] and co-labelled them with nanobodies NbA488 and NbA594. In this case, only two FRET regimes could be determined: i) low or no-FRET ($E_{app} < 0.7$) and ii) moderate FRET efficiency ($0.7 < E_{app} < 0.17$) (Fig. 2e) suggesting that Env in immature viruses adopts at least two conformational states. The addition of saturating concentrations of sCD4$_{D1-D4}$ to immature virions stabilized the low or no-FRET open conformation ($E_{app} < 0.7$) (Fig. 2f), as observed in mature virions, showing that immature particles, although impaired for fusion[7,8,38], retain the ability to adopt an open conformation.

When comparing the HIV Env conformations in unbound mature-enriched and immature HIV-1$_{Gag-GFP\ HXB2\ V4-GFPOPT}$ samples (Fig. 2c, e), the most prevalent conformation in both cases was the one showing intermediate FRET efficiencies. It has been previously reported that unligated mature Env preferentially

adopts a closed conformation[20,39]. Therefore, we hypothesized that these moderate FRET values could represent a closed ground-state conformation, as this conformation would reduce the distance between V4 loops within the Env trimer and thus, labels could be close enough to give intramolecular FRET. In turn, both mature and immature virions in presence of sCD4$_{D1-D4}$ (Fig. 2d, f) showed a predominant low or no-FRET efficiency population, which we could attribute to open Env conformation. Interestingly, high FRET regimes were only observed in the mature-enriched viral sample ($E_{app} > 0.23$), preferentially in the absence of sCD4$_{D1-D4}$ ligand (Fig. 2c). Given that the Env distribution within the viral membrane depends on the maturation state of virions[8], we attributed high FRET regimes to intermolecular Env interactions.

To confirm that high FRET regimes ($E_{app} > 0.23$) observed in the mature-enriched viral sample correspond to intermolecular interactions, HIV-1 pseudo-particles were produced incorporating the GFP$_{OPT}$ in the V1 loop of gp120 Env glycoprotein instead of the V4 loop (HIV-1$_{Gag-GFP\ HXB2\ V1-GFPOPT}$). Note that this specific labelling strategy was also previously tested for fusion[28]. This labelling approach that positions the donor and acceptor fluorophores proximal to the apex of Env when adopting a closed conformation[15], is expected to increase the distance between different Env trimers, thereby minimizing the number of acceptor molecules per donor which would drastically reduce or eliminate the intermolecular FRET[40], if any, between Envs (Supplementary Fig. 3). Viruses with Env labelled with GFP$_{OPT}$ in V1 were exposed to nanobodies coupled to the donor alone (NbA488) or to both, donor and acceptor dipoles (NbA488 and NbA594, respectively). We observed a slight increase in the FRET efficiency in presence of the FRET pair compared to the donor alone condition in both, mature-enriched (Supplementary Fig. 3a, b) and immature viral samples (Supplementary Fig. 3c, d). However, FRET efficiency was not higher than 0.23, as observed in mature-enriched HIV-1$_{Gag-GFP\ HXB2\ V4-GFPOPT}$ viral samples, showing that labelling of the V4 loop in Env is critical to observe Env intermolecular interactions occurring in mature HIV-1 virions.

STED super-resolution microscopy further confirmed that the mature-enriched viral sample contained a higher proportion of virions in which Env was located in a single spot (Supplementary Fig. 4) with an XY (lateral resolution) between 40 and 80 nm, as observed with other labelling strategies utilizing primary and secondary antibodies by others[2,8]. Even if super-resolution STED microscopy allowed to break the diffraction limit and image Env distribution patterns; this resolution was not enough to resolve inter- or intramolecular dynamics. Therefore, only combining FRET-FLIM to study mature and immature HIV-1 particles, allowed us to discriminate intramolecular (open: $E_{app} < 0.12$; closed: $0.12 < E_{app} < 0.23$) from intermolecular ($E_{app} > 0.23$) interactions. Of note, the distinction between intramolecular conformations and intermolecular distribution of Env within the viral membrane in this FRET-FLIM system does not imply that both occur in a mutually exclusive manner within the viral membrane. However, this particular labelling strategy allows detection of intramolecular conformations only when adjacent labelled Env are dispersed enough to prevent intermolecular FRET. These results show that, while transitions between intramolecular conformations occurred in both, mature and immature particles, intermolecular interactions were only observed in mature HIV-1 virions, suggesting that HIV-1 maturation affects Env-Env interactions but not intramolecular Env conformations. These data also show that sCD4 not only stabilized the open Env conformation, but also induced separation of Env molecules as the high FRET regime detected in mature HIV virions was drastically reduced.

**Intermolecular Env clusters are destabilized during the pre-fusion reaction in live T cells.** In order to investigate the intra- and intermolecular dynamics of HIV-1 Env in a physiological context, we studied the sequence of intra- and intermolecular transitions of HIV-1$_{Gag-GFP\ HXB2\ V4-GFPOPT}$ virions labelled with donor (NbA488) and acceptor (NbA594) fluorophores when engaged with MT-4 T cells (Fig. 3a). We examined the time-resolved lifetimes and apparent FRET efficiencies that were simultaneously acquired at a time resolution of 3 s per frame during 5 min (Fig. 3b). The three different $E_{app}$ regimes previously described (low, $E_{app} < 0.12$; intermediate, $0.12 < E_{app} < 0.23$ and high, $E_{app} > 0.23$) were taken as a reference to filter out each of the dwell times coming from individual $E_{app}$ trajectories (Fig. 3b). The three dwell time distributions coming from at least 24 individual HIV-1 virions with a good signal to noise (between 100 and 1000 photons per pixel) were plotted as cumulative distribution functions (CDF) that, in turn, represent the average kinetics of each Env state (Fig. 3c). The average lifetime of each one of the CDF provides quantitative information on the stability of each dynamic state: a shorter CDF average lifetime implies a fast transition, and therefore, a very unstable Env state, and a long CDF average lifetime translates instead in slow Env kinetics and stable Env state. We first analysed the CDF kinetics of mature HIV-1$_{Gag-GFP\ HXB2\ V4-GFPOPT}$ particles in vitro. For the low $E_{app}$ regime, corresponding to the open Env conformation, we observed an average lifetime of $\tau_{(av)} = 84$ s. The intermediate $E_{app}$ regime ($0.12 < E_{app} < 0.23$) CDF kinetics, corresponding to the closed Env conformation gave a $\tau_{(av)} = 154$ s. The long CDF lifetime of this particular closed Env conformation assumes a very stable and predominant state over the open conformation for unbound Env. In turn, the high FRET kinetic regime corresponding to Env-Env interactions ($E_{app} > 0.23$) gave a $\tau_{(av)} = 68$ s.

When the HIV-1$_{Gag-GFP\ HXB2\ V4-GFPOPT}$ virions were engaged with living MT-4 T cells ($n = 20$), both the intramolecular and intermolecular Env landscape drastically changed compared to the previous condition with unliganded Env (Fig. 3c). A delayed and therefore more stable CDF kinetics was found for the populations corresponding to Env open conformation ($\tau_{(av)} = 168$ s). The dynamic behaviour of the Env closed conformation in presence of T cells showed an average lifetime of $\tau_{(av)} = 63$ s, which is faster and therefore less stable than in absence of T cells (Fig. 3c, middle chart); suggesting that Env might have already interacted with CD4 molecules exposed to the cell membrane of MT-4 T cells and hence, inducing an open conformation in the prefusion reaction[9]. Finally, we observed that intermolecular Env interactions when engaged with T cells were destabilized as compared to virions in vitro ($\tau_{(av)} = 53$ s).

This generalized behaviour for the three Env FRET regimes defined above followed a similar tendency with the addition of sCD4$_{D1-D4}$ (Fig. 3d). The open conformation was readily stabilized upon the addition of the HIV-1 soluble receptor ($\tau_{(av)} = 241$ s; Fig. 3d, left chart), as opposed to the closed conformation which was clearly destabilized ($\tau_{(av)} = 75$ s; Fig. 3d, middle chart). Destabilization of Env intermolecular interactions upon addition of sCD4$_{D1-D4}$ was more drastic compared to virions engaged with T cells ($\tau_{(av)} = 16$ s, Fig. 3d, right chart), which might be the result of a higher number of Env molecules binding to its receptor, due to the exposure of virions to saturating concentrations of sCD4$_{D1-D4}$.

We have thus shown that HIV-1 Env open conformation is more stable in virions engaged with T cells compared to cell-free virions. This implies an overall increase in CDF average lifetime for Env open conformation of 84 s ($\Delta_{time} = \tau_{(av)\ (T\ cells)} - \tau_{(av)\ (in\ vitro)} = 168\ s - 84\ s = 84$ s); concomitantly the CDF average lifetime for the Env closed conformation was shorter and more unstable with an overall decrease of 91 s ($\Delta_{time} = 63\ s - 154\ s = -91$ s). Finally, the kinetics of Env dissociation were also favored, giving rise to shorter CDF

average lifetimes and more unstable Env intermolecular interactions with an overall decrease of 15 s ($\Delta_{time} = 53\ s - 68\ s = -15$ s). Overall, these results show that Env intramolecular dynamic states are shifted towards more stable Env open conformation, as opposed to the closed conformation, in sCD4-bound virions or when primed to T cells. Similarly, intermolecular interactions were less frequent under these conditions, suggesting a potential dissociation of adjacent Env molecules into separate trimers upon engagement with CD4 on T cells.

**Disruption of HIV-1 Env dynamics as a mechanism for antibody neutralization.** Next, we examined how the presence of different bNAbs affected Env dynamics when engaged in the prefusion complex with living T cells (Fig. 4). These bNAbs do recognize different Env regions of vulnerability and show selective preferences towards specific Env conformations[25]. PGT145 has been previously reported to recognize the Env apex trimer, stabilizing a closed conformation of HIV-1 Env[13]. B12 binds to an overlapping region of gp120 with the site of CD4 attachment, although as opposed to CD4, is unable to bind the closed conformation of Env. Upon binding, b12 prevents reversion back to the closed state[14], thus stabilizing an intermediate/open conformation[17]. Finally, 10E8 targets a quaternary epitope including lipid and membrane-proximal external region (MPER) contacts[22] stabilizing an open conformation[39]. First, we tested the ability of these antibodies to neutralize HIV-1 infection in TZM-bl reporter cells (Supplementary Fig. 5). We observed a comparable effect in terms of neutralization efficiency of the antibodies in virions bearing the HXB2 WT Env glycoprotein or those harbouring the GFP$_{OPT}$ label in the V4 loop (Supplementary Fig. 5a–c). Results showed a strong neutralization efficiency in the case of b12 and 10E8 antibodies, analogous to the effect induced by exposure of virions to sCD4. Saturating concentrations of PGT145 instead did not significantly affect the viral entry of viruses pseudo-typed with labelled or unlabelled HXB2 envelope, although exposure to PGT145 impaired viral infectivity of HIV-1 virions pseudo-typed with JRFL or NL4-3 (tier 2 and 1 A, respectively[41]) (Supplementary Fig. 5d, e, respectively). These results confirm that labelling of HXB2 in the V4 loop with GFP$_{OPT}$ does not affect viral Env functionality nor epitope accessibility of the ligands included in this study.

HIV-1$_{Gag-GFP\ HXB2\ V4-GFPOPT}$ viruses on the surface of MT-4 T cells were exposed to saturating concentrations (100 μg/mL) of PGT145, b12 and 10E8, and analysed by FRET-FLIM. Our smFRET data indicate that exposure of PGT145 slightly destabilized Env intramolecular dynamics: from $\tau_{(av)} = 168$ s to $\tau_{(av)} = 131$ s in case of the open Env conformation and from $\tau_{(av)} = 63$ s to $\tau_{(av)} = 55$ s in case of the closed Env conformation (Fig. 4a, left and middle chart). We observed a destabilization of the Env open conformation when virions were incubated with T cells in presence of the b12 neutralizing antibody (from $\tau_{(av)} = 168$ s to $\tau_{(av)} = 61$ s). Concomitantly, b12 induced stabilization of the closed Env conformation (from $\tau_{(av)} = 63$ s to $\tau_{(av)} = 125$ s; Fig. 4b, left and middle chart), although not as stable as in the case of virions in vitro ($\tau_{(av)} = 154$ s; Fig. 3c–d, middle panel). In turn, the anti-MPER antibody, 10E8, induced a strong stabilization of the open Env conformation (from $\tau_{(av)} = 168$ s to $\tau_{(av)} = 240$ s) and consistently, the closed conformation kinetics were very similar in the absence or in presence of the antibody (from $\tau_{(av)} = 63$ s to $\tau_{(av)} = 60$ s) (Fig. 4c, left and middle chart). Interestingly, the three antibodies tested induced a drastic destabilization of the Env intermolecular interactions (from $\tau_{(av)} = 53$ s to $\tau_{(av)} = 10$ s, for PGT145; to $\tau_{(av)} = 15$ s, b12; to $\tau_{(av)} = 6$ s, 10E8) (Fig. 4a–c, right chart).

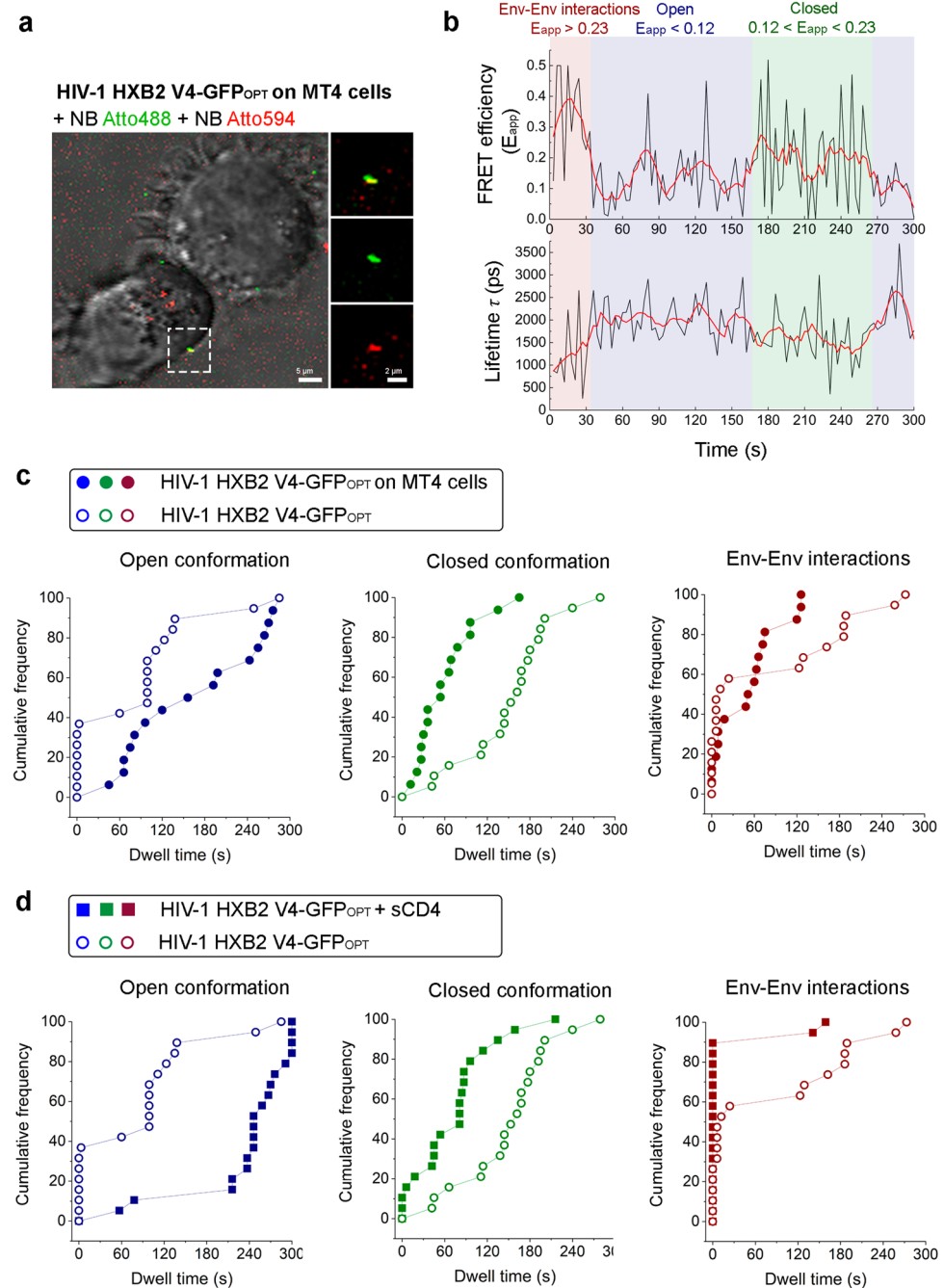

**Fig. 3 HIV-1 Env cluster is destabilized when engaged in the pre-fusion reaction in live T cells. a** Micrograph showing mature HIV-1 HXB2 V4-GFP$_{OPT}$ virions labelled with NbA488 (green) NbA594 (red) engaged in the pre-fusion reaction onto living MT4 T cells (phase contrast). The scale bar image on the left is 5 µm. Magnification of the region contoured with dashed lines is shown on the right. Viral particle showing colocalization between green and red channels is shown on the upper right panel. The middle and bottom-right panels correspond to the same viral particle as observed in green and red channels, respectively. Scale bar magnification is 2 µm. **b** Graphs represent FRET efficiency (E$_{app}$) and lifetime (in ps) traces over time. High FRET efficiency burst (E$_{app}$>0.23), defining intermolecular interactions is depicted in red; intermediate FRET efficiency regime (0.12<E$_{app}$<0.23) assigned to closed Env conformations is depicted in green, and low FRET efficiency bursts (E$_{app}$<0.12) reporting open Env conformations is depicted in blue. Note that high FRET efficiency correlates with low lifetime values and vice versa. **c** Cumulative Distribution Functions (CDF) are plotted for E$_{app}$ single traces obtained from at least (*n* = 20) HIV-1 HXB2 V4-GFP$_{OPT}$ virions in vitro (open dots) and in presence of living T cells (solid dots). Each FRET regime determines the Env conformational state/Env distribution and kinetics. **d** Analysis as in (**c**) of HIV-1 HXB2 V4-GFP$_{OPT}$ virions in vitro (open dots) and in presence of sCD4 (solid squares).

In light of these results, experimentally determined Env-antibody fragment complexes (together with manually positioned GFP labels) and corresponding conformations observed in this study are indicated in Fig. 4d. In our system, intramolecular Env

dynamics were slightly destabilized when incubating HIV-1 virions engaged with T cells in the presence of PGT145, whereas the b12 antibody favours an intermediate intramolecular conformation of Env and, instead, a stable open conformation

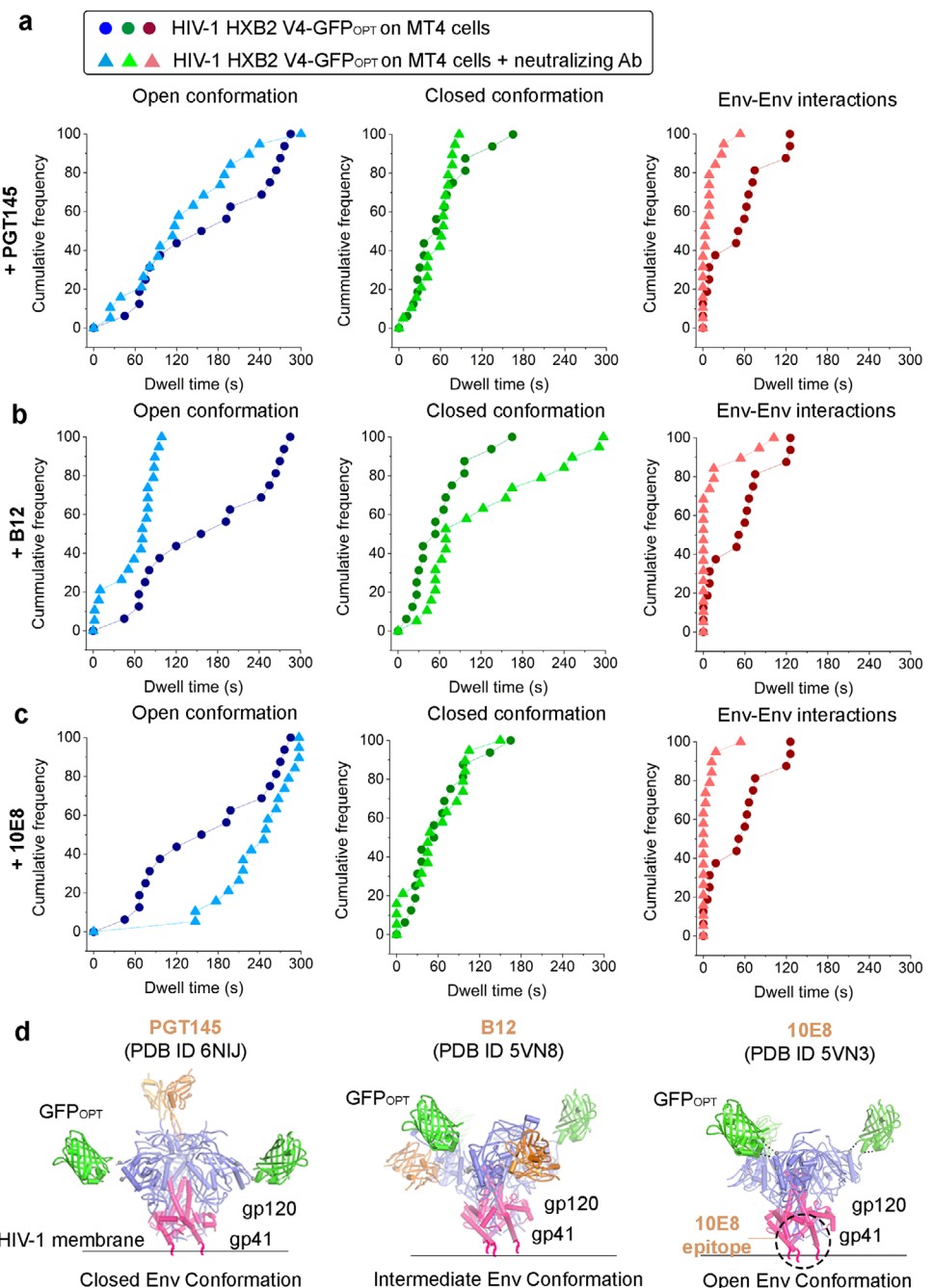

**Fig. 4 HIV-1 Env association is destabilized when exposing virions to neutralizing antibodies in live T cells.** Cumulative Distribution Functions (CDF) are plotted for FRET efficiency traces obtained from at least (n = 20) HIV-1 HXB2 V4-GFP$_{OPT}$ virions in presence of living T cells without (solid dots) or with (solid triangles) neutralizing antibodies PGT145 (**a**), b12 (**b**), and 10E8 (**c**). Each FRET regime (high, E$_{app}$>0.23 in red; intermediate, 0.12<E$_{app}$<0.23, in green; low, E$_{app}$<0.12, in blue) determines the Env conformational state/distribution and kinetics. **d** The HIV-1 Env structures representing closed Env associated with PGT145 (PBD ID 6NIJ, left), b12 (PBD ID 5VN8, middle) or 10E8 (PBD ID 5VN3, right) were modified to illustrate the labelling of the V4 loop with GFP$_{OPT}$.

was observed upon 10E8 addition. Furthermore, these results show that bNAbs, which are known to stabilize intramolecular conformations of Env, strongly impair the intermolecular dynamics of Env by destabilizing and dissociating the Env-Env interactions during the pre-fusion reaction of HIV-1 virions on T cells. Therefore, these results suggest that disruption of Env dynamics is a common mechanism of bNAbs even though each one of them binds to different Env regions. Moreover, these data also point to Env dynamics disruption as an effective and potentially common strategy to inhibit HIV-1 fusion with T cells.

## Discussion

In this study, we investigated HIV-1 Env dynamics using FRET-FLIM imaging. This system allowed us to reconcile dynamics of intra- and intermolecular interactions of Env, which dynamically transits between open and closed conformations and Env-Env association and dissociation in mature, unliganded HIV-1 virions. We have quantitatively shown how Env intermolecular interactions are reduced when primed to live MT-4 T cells. Furthermore, we have shown that three different families of bNAbs, targeting different Env epitopes (PGT145 targets the apex, b12 the CD4-

binding region and 10E8 the MPER region of Env)[25] disrupt Env intra- and intermolecular dynamics when engaged with live T cells. These data were further validated with built-in controls within the same experiments which gave us concomitant no preference upon Env binding to PGT145, an intermediate Env conformation in case of b12 and open Env conformation stabilization when bound to 10E8 or CD4.

The experimental design in this study has been crucial to evaluate intra- and intermolecular interactions of Env in native virions in vitro and engaged with T cells. Key parameters include the time-scale of acquisitions, the labelling strategy and the resolution of dynamic interactions.

Previous results on Env intramolecular dynamics were recovered employing single-molecule FRET combined with Total Internal Reflection Microscopy (TIRF)[19,20,36]. This technology combined with dually labeled Env molecules using short peptides introduced into different gp120 loops provided an outstanding platform to evaluate V1 loop conformational changes[20]. The time-acquisition for their single-molecule FRET experiments was restricted to ~10 s (with 25 frames per second) prior to bleaching of the fluorophores. Although with a great time-resolution, this technique would fail to detect long-time lapse Env dynamics (in the range of seconds and minutes). Importantly, the high power laser needed to collect enough photons for the analysis would induce virus phototoxicity[42], which in turn might affect Env dynamics. Of note, our two-photon FRET-FLIM data were acquired for 5 min with a FLIM time resolution of 3 s, with minimal photobleaching. Moreover, according to our own data presented here, and to a number of biophysical studies[2,8] a longer time-scale might be crucial to detect Env intermolecular conformational dynamics given the slow Env diffusion coefficient in mature HIV-1 particles ($D = 0.002$ $\mu m^2$/sec in mature particles[2]). In this sense, the excitation source utilized in this work has been key to visualize HIV-1 Env dynamics engaged with living T cells. Two-photon excitation provides high three-dimensional contrast and resolution without the need for optical filters (i.e. pinhole or notch filters) in the detection path. This, combined with digital photon counting (HyD) descanned detectors situated very close to a high numerical aperture objective, gave a high signal-to-noise ratio. Since two-photon excitation is naturally confocal[43], only the viruses engaged with MT-4 T cells were imaged and all emission photons gave a valuable signal, with reduced phototoxicity, allowing longer acquisitions times whilst conserving resolution and high contrast. Lastly, two-photon excitation also provided a localized excitation where all emission photons constitute a useful signal that contributed to our rapid FLIM acquisitions.

The labelling strategy when performing FRET experiments is also crucial. A labelling approach strictly circumscribed to Env could be a restraint. In this scenario, one could not guarantee that all particles analyzed could be bona fide HIV-1 virions with their corresponding capsids. Here, we tagged the HIV-1 capsid (Gag-GFP) and the gp120 V4 domain of Env with a super folder GFP (GFP$_{OPT}$)[28]. On top of that, we employed labelled nanobodies[9] that specifically bind to GFP (in our system, only GFP$_{OPT}$ is accessible to nanobodies) that could be accurately detected. At least 5 amino acid residues linking GFP$_{OPT}$ with the V4 and V1 loops of gp120 allow free, or random rotation of the fluorophores, thus, minimizing problems related to the relative dipole-dipole orientation (and here, one could approximate $K^2 = 2/3$). Under these circumstances, FRET interpretation is restricted to protein folding and protein-protein interactions. (Fig. 5).

Choosing the label location within the Env glycoprotein is not trivial. To resolve both, intra- and intermolecular interactions, the V4 loop of the gp120 was selected, as the side location of this residue facilitates FRET to occur between different Env trimers without altering Env functionality (Fig. 1e–g and Supplementary

Fig. 5a-c)[28]. Also, in our system, all Env proteins within HIV-1 virions were labelled with GFP$_{OPT}$, increasing the presence of multiple acceptors per donor molecule in conditions of Env cluster formation and thus, facilitating an increased energy transfer and an additional shortening of the donor lifetime[40].

In previous reports, Env cluster characterization has been performed applying techniques such as STED[2,8,44,45], 3D STORM[46] or Cryo-ET[47,48]. While these techniques offer an incredibly high spatial resolution, they are unable to resolve Env conformational dynamics. Here, we employed a FRET-FLIM approach as a molecular ruler for the study of different Env conformation transitions on the nanoscale (~2–10 nm). This technique allowed us to detect modulations of Env trimer interactions in real-time when binding to neutralizing antibodies or upon CD4-receptor binding of HIV-1 virions onto T cells, which were previously unappreciated[8,44]. However, using our FRET-FLIM system we are unable to discern whether disruption of Env-Env interactions (distance above 10 nm) upon CD4 binding or by neutralizing antibodies results in a complete redistribution of Env within the viral envelope or instead, it is a subtle intermolecular separation which cannot be resolved with STED microscopy, in which spatial resolution is limited to ~60 nm (Supplementary Fig. 4). Nevertheless, detection of Env disaggregation does not necessarily exclude the importance of Env clustering to maximize the changes for Env-CD4 interaction, nor imply that only one Env molecule is required for Env fusion. Our previous work[9] showed that HXB2 Env required at least two Env during the prefusion reaction whilst JRFL Env needed at least one. This was further validated in a recent study[3] were the HIV-1 spokes were imaged using CryoET when engaged with TZM-bl cells. This work further confirms that 1 or 2 Envs are needed during the first stages of the fusion reaction, and this vision would confirm that the previous Env cluster dissociates right at the moment of Env-CD4 engagement.

In previous studies, HIV-1 Env has been described to exist in at least three intramolecular conformational states by using smFRET approaches[18–20,36]. Unligated Env dynamically transits between different intramolecular conformations, although with a preference to pre-triggered, closed conformation (state 1). Binding to CD4 would first induce an asymmetric opening of Env (state 2) that ultimately leads to binding to its coreceptor CXCR4 or CCR5 (state 3) for subsequent fusion with the host membrane. Ensemble analysis of E$_{app}$ in our FRET-FLIM system, resolved two different intramolecular conformations. We designed them as open and closed conformations. Open conformation showed the lowest FRET efficiency and was stabilized upon binding to sCD4$_{D1-D4}$ in mature and immature virions. The concentration used in this assay (10 µg/mL) has been associated to three-CD4-bound conformation[49], hence we might relate this conformation with previously described in the literature as state 3. Closed conformation was characterized in our approach by moderate FRET efficiency regimes and was predominant in unliganded mature virions. Therefore, the closed Env conformation in our system could relate to previously assigned as state 1 conformation. The intermediate opening of Env was not clearly resolved by our ensemble E$_{app}$ analysis. However, we observed subtle differences in the degree of stabilization of the open conformation when comparing E$_{app}$ kinetics in mature HIV-1 in the absence or presence of T cells or in the absence or presence of the bNAbs b12 or 10E8, suggesting the existence of intermediate open states with different degrees of stability as previously observed[19,39].

When comparing multiparameter plots of E$_{app}$ and lifetime of mature-enriched vs immature viral samples, we observed that high FRET efficiency regimes relate to intermolecular interactions of Env, as they were only observed in mature-enriched viral samples labelled at the V4 loop of gp120. When analyzing E$_{app}$

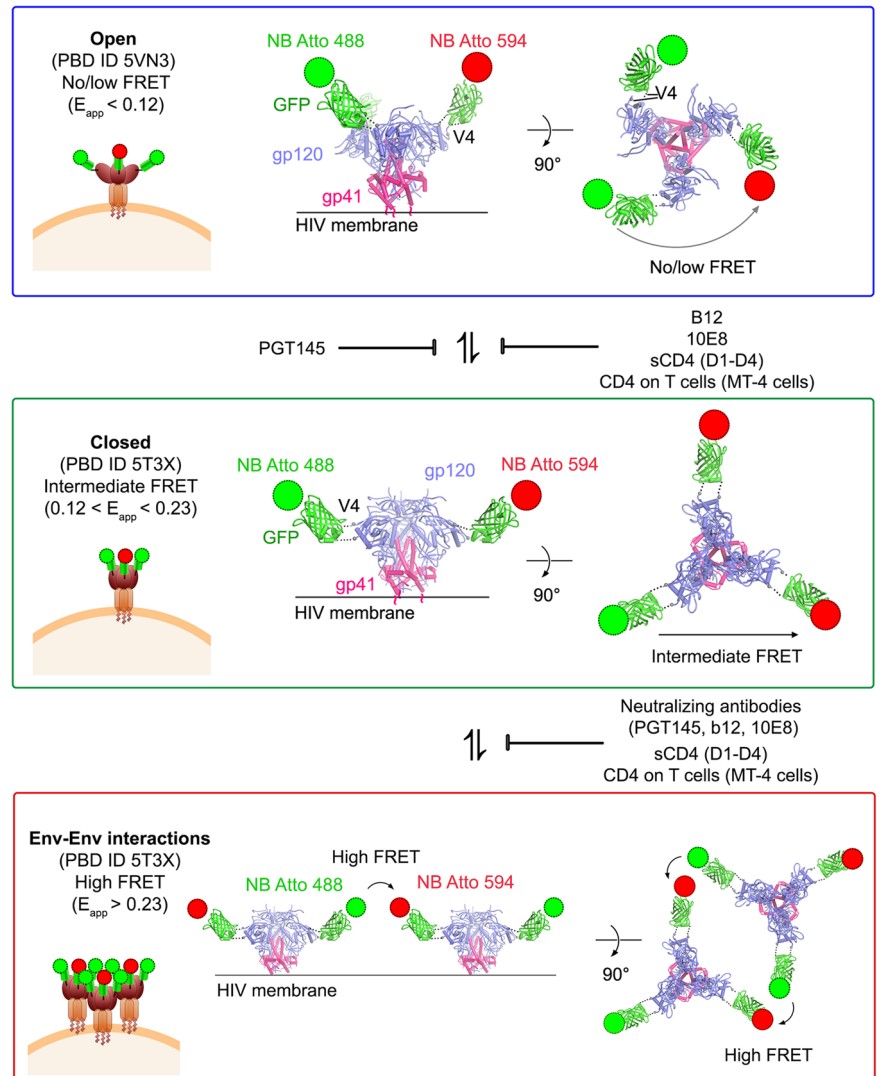

**Fig. 5 HIV-1 Env transitions between intra- and intermolecular conformations are disrupted by CD4-binding and bNAbs.** Schematic representation of the transitions between different intra- (open, showing low/no FRET, upper panel; closed, showing moderate FRET, middle panel) and intermolecular (cluster, showing high FRET bottom panel) Env conformations observed by FRET-FLIM. Receptor binding (sCD4$_{D1-D2}$ or CD4 on T cells) and binding of b12 and 10E8 bNAbs stabilize and intermediate or open conformation, whereas PGT145 favors a closed conformation. Engagement of Env with CD4 or with any of bNAbs tested increases the distance between Env molecules, inducing Env-Env dissociation.

time-resolved tracks from mature HIV-1 particles in vitro, and also primed to T cells, the transition towards Env clusters always occurred via the Env closed conformation, and never from an Env open conformation (Fig. 5). The labelling strategy used in this study allow us to discriminate between different Env intramolecular conformations when the intermolecular distance between adjacent labelled Env proteins is above FRET resolution (>10 nm). However, intramolecular dynamics cannot be resolved when high FRET regimes (corresponding to Env-Env interactions) are observed. The addition of stabilizing mutations in Env would help to decipher which intramolecular Env conformation is preferably adopted during cluster formation and whether there is a more favourable Env conformation triggering cluster formation.

An important consideration to study Env conformational dynamics is the temperature at which experiments are performed, as the HIV-1 fusion reaction is known to be temperature dependent[50]. It has previously been shown, however, that receptor priming can occur at 4 °C and that temperature-arrested state (TAS) can generate a stable intermediate that is kinetically

predisposed to complete the fusion event leading to infection[50]. In our case, all experiments were carried out at room temperature as described in the previous studies[2,20] and therefore receptor priming and the first steps of the fusion reaction can proceed. Future studies performed at physiological fusion-permissive temperature would help to relate these findings with productive HIV-1 fusion at 37 °C.

Neutralizing antibodies included in this study were selected for their ability to recognize different epitopes in Env. HIV-1 Env is able to transition between intramolecular conformations prior to receptor binding, which results in different susceptibility to antibody neutralization[25,41]. Hence, HIV-1 Env trimers of strains belonging to tier 1 A (HXB2 or NL4-3) are more prone to adopt an open conformation, whereas tiers 2 (JRFL) and 3 to a closed conformation[41]. Consistently, our neutralization assays showed that antibodies targeting HIV-1 Env presented different abilities to neutralize depending on the viral strain tested (Supplementary Fig. 5). Importantly, labelling the V4 loop of gp120 HXB2 did not affect the neutralization profile observed in the WT HXB2 pseudo-typed viruses, confirming that Env dynamics are

preserved in spite of V4-GFP$_{OPT}$ labelling. FRET-FLIM analyses showed that Env intramolecular dynamics were slightly affected when exposed to the antibody PGT145 and, accordingly, PGT145 showed low neutralization efficiency in TZM-bl cells. This could be explained by a relatively low frequency of the HXB2 Env to adopt a closed conformation compared to other HIV-1 Env strains, which would prevent recognition by PGT145. Hence, the addition of b12 or 10E8 antibodies, which stabilize an intermediate/open conformation, strongly neutralized infection of HXB2 pseudo-typed viruses. The distinct ability of PGT145 to neutralize HIV-1 strains could also be explained by the glycosylation at the V2 loop, which modulates sensitivity to this antibody[51].

Even though neutralizing antibodies affected differently intramolecular conformations of Env, we observed a common effect relative to intermolecular dynamics. Indeed, the three antibodies tested (PGT145, b12, and 10E8) disrupted Env-Env interactions. Consistent with the previous studies[2], our results showed that intermolecular dynamics were affected in immature HIV-1 particles, but importantly, we still observed transitions between intramolecular dynamics. Specific inhibition of the HIV-1 protease with SQV dramatically impaired viral infectivity. Impairment of the fusion activity in immature HIV-1 particles has been related to the CT of gp41, which remained locked in an inactive state as a consequence of defective processing of the HIV-1 Gag precursor by the viral protease[6]. These results suggest that blocking Env-Env interactions prior to receptor binding could prevent optimal pre-fusion engagement with the host cell thereby blocking membrane fusion. Altogether, our results show that the maturation state of HIV-1 virions affects intermolecular but not intramolecular dynamics of Env. Neutralizing antibodies disrupt trimer dynamics concomitant with a destabilization of Env-Env interactions, suggesting that both, intramolecular dynamics and intermolecular Env-Env association play a synergic role allowing virus fusion.

Overall, our data clearly shows that intramolecular dynamics and intermolecular Env association and dissociation is a key factor in mature HIV-1 particles. Moreover, the mechanism of masking different Env regions by neutralizing antibodies might also be related to the intermolecular dynamics of Env and immune evasion. Taken together, this work suggests that destabilizing Env dynamics could represent a common strategy to arrest and inhibit viral fusion machines.

## Methods

**Cell Culture**. TZM-bl cells used in functional assays (kindly provided by Dr Quentin Sattentau, University of Oxford, United Kingdom) were cultured in Dulbecco's Modified Eagle Medium (DMEM) supplemented with GlutaMAX™, 10% FBS and 1% PS (complete DMEM). Lenti-X™ 293 T cells used for pseudovirus production (Takara Bio, Clontech, Saint Germain en Laye, France) were grown using complete Dulbecco's Modified Eagle Medium F-12 (DMEM F-12) (Thermo Fisher Waltham, MA, USA), supplemented with 10% fetal bovine serum (FBS), 1% penicillin-streptomycin (PS), and 1% L-glutamine. MT-4 T cells (provided by Dr Alex Compton, NCI Center for Cancer Research, Frederick, MD, USA) were cultured in RPMI 1640 medium containing 10% FBS, 1% PS and 1% L-glutamine. Cells were maintained in a 37 °C incubator supplied with 5% CO$_2$. For FRET-FLIM experiments, MT-4 T cells were cultured in PBS 1x buffer containing 2% FBS and 15 mM HEPES.

**Reagents and antibodies**. Nanoboosters (Chromotek, Germany) targeting the GFP$_{OPT}$ of labeled Env GFP_Booster_Atto488 and/or RFP_Booster_Atto594 (ChromoTek GmbH, Planegg, Germany) were used in 2.5 µg/mL final concentration. Human soluble CD4 recombinant protein (sCD4$_{D1-D4}$; Cat.No:4615, NIH AIDS reagent program) and broad neutralizing antibodies: anti-HIV-1 gp120 monoclonal, PGT145 (Cat. No: 12703, NIH AIDS reagent program); anti-HIV-1 gp41 monoclonal, 10E8 (Cat. No: 12294, NIH AIDS reagent program) and anti-HIV-1 gp120 monoclonal, b12 (Cat. No: AB011, Polymun Scientific, Klosterneuburg, Austria), were used in FRET-FLIM experiments at 10 µg/mL final concentration in case of sCD4 and 100 µg/mL for experiments with antibodies.

**Plasmid constructs**. The pR8ΔEnv plasmid (encoding HIV-1 genome harbouring a deletion within Env), pcRev, NL4-3 Gag-iGFPΔEnv were kindly provided by Dr Greg Melikyan (Emory University, Atlanta, GA, USA). The plasmids encoding the HXB2 gp120 V4 and V1 labeled with GFP$_{OPT}$ were a kind gift from Dr Zene Matsuda (Institute of Biophysics, Chinese Academy of Sciences, China). The plasmid encoding the JRFL Env was a kind gift from Dr James Binley (Torrey Pines Institute for Molecular Studies, USA) and NL4-3 Env coding plasmid was provided by Dr Alex Compton (NIH, USA)

**Virus production**. Gag-GFP-containing, HXB2 V4-GFP$_{OPT}$ and HXB2 V1-GFP$_{OPT}$ pseudotyped viral particles were produced via transfection of 60–70% confluent Lenti-X™ 293 T cells seeded in T175 flasks. DNA plasmids were transfected into Lenti-X™ 293 T cells using GeneJuice® (Novagen, Waltford, UK) according to the manufacturer's protocol. Specifically, cells were transfected with 2 µg pR8ΔEnv, 1 µg pcRev, 3 µg of NL4-3 Gag-iGFPΔEnv and 3 µg of the appropriate viral envelope. All transfection mixtures were then added to cells supplemented with complete DMEM F12, upon which time they were incubated in a 37 °C, 5% CO$_2$ incubator. 12 h post-transfection, the medium was replaced with fresh, phenol-red free, complete DMEM F12 after washing with PBS. In the case of immature HIV-1 pseudovirus production, complete DMEM F-12 was supplemented with 300 nM of the HIV-1 protease inhibitor Saquinavir mesylate (Sigma-Aldrich, St. Louis, MO, USA). 72 h post-transfection, the supernatant containing virus particles was harvested and filtered with a 0.45 µm syringe filter (Sartorius Stedim Biotech). Filtered viral supernatants were concentrated 100 times using Lenti-X™ Concentrator (Takara Bio, Clontech, Saint Germain en Laye, France) and resuspended in phenol red-free medium, FluoroBrite DMEM (Thermo Fisher, Waltham, MA, USA), aliquoted and stored at −80 °C.

**Infectivity assays**. TZM-bl cells were used for infectivity assays. This cell line stably expresses CD4 and co-receptors CCR5 and CXCR4, required for HIV-1 infection. Moreover, they have been engineered to express the β-galactosidase gene under the control of the HIV-1 LTR promoter. Therefore, infected cells will produce the β-galactosidase enzyme capable of hydrolyze the X-gal substrate when added onto cells. This reaction will produce a characteristic blue color in infected cells that can be easily identified by transmission microscopy. One day prior to infection, $2 \times 10^4$ TZM-bl cells were seeded per well in a 96-well plate (so that cells reached ~90–100% confluency the day of the X-gal assay). Serial dilutions of the equivalent amount of physical viral particles, judged from the O.D. measurements of p24 by ELISA (Cat: SEK11695, SinoBiological, UK), were diluted in complete DMEM and added onto to cells and incubated for a further 48 h. Infected cells were then fixed using 2% paraformaldehyde (PFA). After fixation cells were washed with PBS and incubated with X-gal solution (5 mM K$_3$[Fe(CN)$_6$], 5 mM K$_4$[Fe(CN)$_6$], 2 mM MgCl$_2$, 1 mg/mL X-gal in PBS) at 37 °C in the dark for 2 h. The X-gal solution was then replaced by PBS and cells were imaged using a Leica DMi8 microscope, Leica Microsystems (Manheim, Germany) equipped with a 10× objective. Images of each condition were obtained after merging 25 tiles. Images were analyzed using the ImageJ software (https://imagej.nih.gov/ij/). A positive signal from infected cells was highlighted to quantify the percentage of area occupied by infected cells. Similar area and threshold settings were applied to analyze each condition and the results were plotted using GraphPad Prism 9.1.0 software.

**Neutralization assays**. One day prior to infection, $10^4$ TZM-bl cells were seeded per well in a 96-well plate. On the day of the assay, HIV-1 viruses were diluted in the cold in the absence or presence (1 µg/mL or 100 µg/mL, as indicated in the graph) of neutralizing antibodies or sCD4 in Fluorobrite DMEM supplemented with 2% FBS and 1% PS. The different reaction mixes were incubated during 1 h at RT in gentle agitation prior to an addition onto TZM-bl cells. Viruses were allowed to infect cells during 2 h at 37 °C before replacing the inoculum with complete DMEM. Cells were incubated for a further 48 h at 37 °C. A similar protocol used for infectivity assays (see above) was applied to quantify the extent of neutralizing activity of antibodies or sCD4.

**Sample preparation for FRET-FLIM analyses**. HIV-1 viruses pseudotyped with labelled HXB2 Env and harbouring Gag-GFP were diluted in PBS 1×, 2% FBS buffer and plated onto a micro-slide (Cat.No: 81826, Ibidi, Gräfelfing, Germany) by centrifugation at 2100 g, 4 ºC during 20 min. Unbound viruses were removed and media replaced by 2.5 µg/mL of nanoboosters (NbA488 and/or Nb594) in presence or absence of 10 µg/mL sCD4$_{D1-D4}$ or 100 µg/mL concentration of bNAbs in PBS 1×, 2% FBS at 20 µl final volume. The sample was then incubated for 1 h at room temperature (RT) before image acquisition. All incubations were done in the dark to preserve the fluorophore staining.

**Single virus tracking**. MT-4 T cells were added onto surface-bound viruses in a final volume of 20 µl. Cells were spun for 10 min at 600 g in a refrigerated centrifuge so that the HIV-1 particles could engage with the cells, without initiating the prefusion reaction. The observation micro-slides were then put under the microscope and the cold medium was immediately replaced by a medium at RT right at the moment when we started the imaging acquisition procedure. We employed the

two-photon SP8 X SMD DIVE FALCON confocal microscope (also described below) equipped with a dark incubator chamber for the frame.

Virus tracking was performed with both ImageJ plugin Spot tracker and the 64-bit software module from Imaris (BitPlane, Zurich, Switzerland), using an auto-regressive algorithm. Tracking provided quantitative information regarding the mean fluorescence intensities of the HIV-1 HXB2-GFP$_{OPT}$ with NbA488 (donor) collected in the non-descanned HyD1, the sensitized emission of the donor in the presence and absence of the acceptor HIV-1 HXB2-GFP$_{OPT}$ NbA488 (donor) and NbA594 (acceptor) was recovered in the second non-descanned HyD2. Tracking of individual particles both in vitro and when engaged in MT-4 T cells also provided FLIM values for the donor, particle's instantaneous velocity, trajectory and the mean square displacements (MSD).

**Fluorescence lifetime imaging microscopy (FLIM).** HIV-1 labeled virions and live MT-4 T cells exposed to HIV-1 particles were imaged using a DIVE SP8–X-SMD FALCON Leica microscope, Leica Microsystems (Manheim, Germany). Both, HIV-1 virions and MT-4 T cells of interest were selected under a 100x/1.4 oil immersion objective corrected for infra-red light (IR). HIV-1 labeled virions were excited using a two-photon femtosecond pulsed laser tuned at 950 nm and 80 Mhz. The FALCON module was coupled with single-photon counting electronics for rapid FLIM (Leica Microsystems) with virtual gating set at 97 ps. Green emission photons were subsequently detected by three hybrid non-descanned external detectors in photon counting mode with emission filters set at 500–550 nm, 600–650 nm and a long pass starting at 700 nm for the third HyD detector. Stacks of 100 images of time-resolved data acquired at 3 s each for 5 min were acquired for all experiments. HIV-1 HXB2-GFP$_{OPT}$ Nb594 (acceptor) particles were tested to avoid the possibility of cross-excitation of the acceptor (Nb594) with the two-photon laser tuned at 950 nm. No photons were detected in the acceptor channel with the power set at 10% of the laser power (Spectra-Physics, UK). Leica software (LAS X) was employed to produce the phasor plots (Leica Microsystems, Mannheim, Germany). ImageJ (https://imagej.nih.gov/ij/) and Originlab (North-hampton, USA) were employed to produce the multiparameter two-dimensional graphs and probability kernel maps comparing the apparent FRET efficiency (calculated as described below) with average lifetime data (given in picoseconds per pixel) and recovered with LAS X and previously treated with ImageJ to remove the noise.

Both, photon counting images for the donor (HyD1) and the sensitized FRET emission and FLIM micrographs simultaneously acquired by the two-photon SP8 DIVE FALCON system using the same microscopy settings were background subtracted, to get rid of the scatter photons and white noise recovered by each HyD channel. After this, each single virus was profiled utilizing a mask that only contained the signal coming from each individual particle (in vitro or engaged with non-labeled T cells) and non-attributed-numbers for the background. Both, the time-resolved intensity and average lifetime values for each channel were obtained together with the average intensity values (in photons and not grey values). The average number of photons and average lifetime per channel for each profiled virus was obtained ($n > 100$ particles for in vitro and $n > 65$ for live-cell imaging) and plotted as a multiparameter plot and phasor plot for each condition. Individual traces for each condition were also recovered following this procedure.

**FRET and FLIM image analysis.** The HXB2 Env was fused to genetically encoded GFP$_{OPT}$ that in turn was labeled by nanoboosters (NbA488, playing the role of the donor and Nb594, playing the role of the acceptor), and the molecular dynamics of Env in question was then inferred by FRET between the fluorophores. The efficiency with which Förster-type energy transfer occurs in given by the next equations:

The FRET efficiency (E) can be calculated as the proportion of photons absorbed in the donor versus the excitation transferred to the acceptor

$$E = \frac{k_t}{k_t + k_D} \qquad (1)$$

Where $k_D$ is the sum of all relaxation pathways and $k_T$ is the transfer rate.

Experimentally we calculated E$_{app}$ pixel by pixel utilizing the next equation

$$E_{app} = \frac{I^{sens}}{I^D} - Bkr \qquad (2)$$

The signal of the laser pulse and delayed photon arrivals were rapidly digitalized at high speed with a temporal resolution per channel of 97 ps, allowing very rapid FLIM acquisitions (1–3 s per FLIM image). Pixel by pixel images with their corresponding background-subtracted average lifetime images where directly provided by the Leica software LAS X with the FALCON module. Single photons coming from the donor/s (HIV-1 HXB2 V4 or V1-GFP$_{OPT}$ labelled with NbA488 (donor) in the presence and absence of NbA594 (acceptor)) were detected in the non-descanned HyD detector. FLIM analysis was performed applying the non-fitting phasor plot approach fully integrated in the LAS X software.

**Intramolecular and intermolecular dynamics model.** The average time-resolved lifetimes and apparent FRET efficiencies that were simultaneously acquired at a time resolution of 3 s per FLIM image during 5 min, coming from individual HIV-1 viruses were filtered using the next criteria (for each condition). The three different

E$_{app}$ regimes previously described (high, E$_{app}$ > 0.23, intermediate 0.12<E$_{app}$<0.23 and low E$_{app}$ < 0.12) were taken as a reference to select each of the dwell times coming from individual E$_{app}$ trajectories. The three dwell time distributions coming from at least 20 individual HIV-1 viruses per condition; were plotted as cumulative distribution functions (CDF). Only HIV-1 particles with a good signal to noise (between 100 and 1000 photons per pixel) were selected for all conditions. The average lifetime was calculated for each CDF curves utilizing the next equation:

$$\tau_{av} = \frac{\sum_i \frac{100}{n} * t_i}{100} \qquad (3)$$

Where $n$ is the number of points and $t$ the corresponding dwell time for each CDF data point.

**STED microscopy.** Super-resolution of HIV-1 Env (Gag-GFP,HXB2-GFP$_{OPT}$) exposed to nanobodies labelled with Atto 594 (NbA594) was performed using the Leica SP8 STED 3X microscope (Mannheim, Germany) equipped with a 100×/1.4 oil immersion STED objective. STED images for Env and confocal for GFP-Gag were acquired sequentially for each channel using 594 and 488 nm lines (with a pulsed white laser tuned at 40 Mhz). The NbA594 signal was depleted with a doughnut-shaped 775 laser. Under these conditions around 40 nm lateral resolution was achieved for HXB2-GFPopt/NbA594. STED images for the red channel were obtained employing 1 Airy unity. 1.3 us/pixel and a pixel size of 20 nm. Analysis of the STED images were carried out with ImageJ software (https://imagej.nih.gov/ij/). Env distribution was assessed manually, evaluating the Env distribution profile as shown in Supplementary Fig. 4 associated with Gag-GFP positive particles.

**Structural models and analysis.** A model for Intra and intermolecular dynamics of the HXB2 Env labeled with GFP$_{OPT}$ and nanoboosters was built using the following structures: ligand-free HIV-1 Env mimic (BG505 SOSIP.664) (PDB: 4ZMJ[11]), HIV-1 Env mimic (B41 SOSIP.664) in complex with the ectodomain of CD4 (PDB: 5VN3[14]). Models were generated in Chimera, Coot, and PyMOL (https://pymol.org).

**Statistics and reproducibility.** For FRET-FLIM analyses, calculation of multi-parameter 2D kernel density plots and t(1/2) of CDF from E$_{app}$ traces was performed using Originlab software (Northhampton, USA). Mean and STD from infectivity assays and assessment of maturation efficiency were calculated using GraphPad Prism 9.1.0 software. The sample size and the number of experiment replicate performed are specified in each figure legend.

**Reporting summary.** Further information on research design is available in the Nature Research Reporting Summary linked to this article.

## Data availability
The data supporting the findings of this study is available at osf.io/2gjc9[52].

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

## Acknowledgements

We thank Zene Matsuda for the kind gift of HIV-1 Env labeled plasmids. We thank Leica Microsystems for technological support. We also thank Dr Chris Lagerholm and the WIMM imaging facilities for technical support in STED imaging. We thank the Padilla-Parra lab for valuable discussions and criticism of the paper. This work has been supported by the European Research Council (ERC-2019-CoG-863869 FUSION to S.P.-P.) and the Wellcome Trust Core Award (203141).

## Author contributions

Conceptualization, I.C.-A. and S.P.-P.; methodology, I.C.-A., T.M. and S.P.-P.; validation, I.C.-A. and S.P.-P.; formal analysis, I.C.-A. T.M and S-P-P; investigation, I.C.-A, T.M. and S.P.-P.; resources, S.P.-P.; data curation, I.C.-A. and S.P.-P.; writing-original draft preparation, S.P.-P.; writing—review and editing, I.C.-A., T.M. and S.P.-P.; visualization, I.C.-A. T.M and S.P.-P.; supervision, S.P.-P.; project administration, S.P.-P.; funding acquisition, S.P.-P. All authors have read and agreed to the published version of the paper.

## Competing interests

The authors declare no competing interests.
