## [Transparent Peer Review File · Communications Biology]

Reviewers' comments:

Reviewer #1 (Remarks to the Author):

Carlou-Andres and colleagues combined in their study single molecule bleach and FRET-FLIM to characterize the dynamics of the HIV-1 Env trimer. The authors demonstrate a clustering effect of Env at the surface of viral particles and show that these clusters are disrupted by CD4 and neutralizing antibodies.

The topic of the study is clearly of high interest and I appreciate the use of alternative methods to monitor the structural dynamics of the Env trimer. However, in its current state the manuscript raises many questions that need to be solved. The validity of the experimental approach is not clear. Advantages and pitfalls of single molecule FRET must be laid out explicitly and the experimental set up verified.

The authors should also put more effort to make the interpretation of their results accessible to non-FRET experts. The result section is a heavy read even for the most dedicated reader. Figures are very detailed and sometimes confusing. The manuscript would benefit from figures that summarize and shifting of original data to the supplement. Figures should be redesigned to use colors that are easy to distinguish, have spacing between sub-panels and include accurate legends.

Specific points:

1. The authors fail to clearly define and distinguish what they term clustered, open and closed conformations. The term conformation is seemingly arbitrary used throughout the paper for both intra- and intermolecular interactions. Judging from text and figures one gets the impression as if clusters and open/closed conformations are mutually exclusive. However, this is clearly not the case and needs to be corrected throughout the manuscript and figures. The authors need to use more distinct terms to clarify the differences between inter- and intramolecular dynamics. While it is fitting with the FRET data to describe this as a sequential process, the in vivo situation calls for a more clear distinction between closed/open Env and cluster association/dissociation.
2. The title should not refer to "structure" as the work solely focuses on dynamics measurements.
3. Line 50: "that exposes otherwise hidden, more conserved epitopes, increasing susceptibility for antibody recognition " This is not clear. More conserved than what? bnAb epitopes on closed trimers are also highly conserved.
4. Line 52: "intramolecular structure": should this rather be "intramolecular interactions"?
5. The introduction is too brief especially considering the broad audience of Nature Communications. Much of the information on previous results and state of research included in the discussion, would have been helpful in the introduction. The introduction needs information on antibodies and their dependence on different conformations, the influence of the maturation process on Env and references to Cryo-ET/EM studies of closed and open Envs.
6. Why are different strains used in different experiments? The choice seems arbitrary.
7. GFP is quite big and to some extent flexible (see also discussion line 427 etc). Since the FRET-signal depends on the distance, this flexibility may lower the resolution between conformational states that can be distinguished.
8. The authors need to show that GFP insertion is not influencing Env conformation /opening by itself. The authors reference Nakane et al. who demonstrated no influence on expression level and fusogenicity. However, this needs to be shown for the current Envs. Also the exposure of CD4i and V3 epitopes may be influenced by GFP insertion. The exposure of these and other Ab epitopes must therefore be probed experimentally and stability of the GFP-Env compared with wild type.
9. The authors analyze their "mature virion prep" using a permeabilization strategy that allows to judge that 40% of the "mature prep" are actually mature virions. Considering that the preps are not fully mature, the preps should be rather referred to as "mature enriched". The corresponding figure S1B does not show any error bars, therefore it is unclear in how far this percentage varies between preps. Also, the percentage of maturation could be time- and temperature-dependent. Was this controlled for?
10. The authors compare mature vs immature virion preps are compared. It is unclear from these

experiments if the “mature preps” always contain mature and immature virions. Judging from Figure 1 and Figure S2 these seems not to be the case, there seems to be no signal from immature virions. How can these be? The authors need to explain how they select solely mature virions from their prep that only contains 40% mature. As permeabilization appears not to have been done in parallel to the FRET experiments to quantify the % mature virions, the authors need to clarify how maturation stage was defined in the experiments and controlled for.

11. Why is the FRET-background assessed only with the donor and not with the acceptor (Figure S2)? Should not capture of the FRET-signal be done at the acceptor wavelength (longer than the donor)? It needs to be shown whether or not GFPopt in Env causes background FRET by transmitting to the acceptor.

12. In the discussion (lines 423-425) the authors state that mature and immature particles can be discriminated by GFP-lifetime- How was this assessed? It is generally unclear why a difference in the GFP lifetime between mature and immature particles would exist. The authors explain that the lifetime of the signal is reduced in Fig 1D due to the FRET. In Fig. 1E (left) they also distinguish some FRET signal in the intermediate range, but here the lifetime appears longer (compared to Fig S2B, right panel)?

13. The FRET results with Envs containing GFPopt in V1 stick out: Here the donor and acceptor should in theory be located more closely together (at least in the closed state) compared to labelling on V4. This should lead to a stronger FRET compared to the labelling on V4. But this is not seen in the data. The authors need to explain this.

14. In the experiments with b12 the authors first incubate the viral particles with sCD4, then fix, and add the labelled b12. As the authors state themselves (lines 341-344), b12 is unable to bind the closed conformation and itself stabilizes an intermediate/open state. Since the cells are fixed, b12 should not be able to induce opening and residual binding should be weak and suffer from an increased off-rate which may lead to an overestimation of photobleaching events. These aspects should be considered in the interpretation of the results. The authors further state that they use a saturating sCD4 concentration but still use the CD4bs Ab b12 as a secondary stain. Since substantial competition should occur, the authors need to explain the rationale of this setup and how the interpret measurements more clearly.

15. In the experiments with bnAbs (Figure 3 and 4) the label for photobleaching is attached to the bnAbs. How is the labeling and photobleaching done for the reference in the absence of antibody (since JR-FL and NL4-3 do not contain GFPopt the labelled nanobody cannot be used instead)?

16. The authors conclude that Env cluster formation requires a closed conformation. However, even in the presence of PGT145 cluster formation is apparently reduced. This should be discussed.

17. In Figure 1D and Figures 5 and 6, the authors look at the FRET signal below 0.12 (attributed to the open state). However they defined in Figure S2A the threshold for background at 0.1. How is the signal below 0.12 accurately quantified?

18. Line 125: A saturating concentration of sCD4 leads to an increase of the no-FRET status. Is this also influenced by shedding of Env? At saturating sCD4 doses this should happen to some degree? At 10ug/ml CD4 shedding of HxB2 Env would be considered highly likely (Figures 1 and 2).

Minor comments:

19. Line 260: It would be important to bring the effects of cluster disruption into context with the binding/neutralization capacity of the respective antibodies and the used isolates. Antibody concentration used need to be stated.

20. Line 269: It is confusing that both cluster formation and intramolecular dynamics are termed “conformation”. Change to cluster formation?

21. Line 320: This sentence is unclear

22. Line 319: potential shedding influence needs to be discussed

23. Line 327: the differentiation into open, closed and clustered as separate conformations is unclear. Why can Env in open conformation not form clusters?

24. Line 346: “PGT145 exhibits a tendency to destabilize the open Env”. The authors state this to explain their FRET finding. However this statement implicates that PGT145 binds and disturbs open Env. Since PGT145 binds closed Env a predominant action on open Envs is not likely. This needs to be clarified.

25. The two result sections describing bnAb effects should be merged
26. The discussion lacks an interpretation on the effect of bnAbs on cluster formation: It is not fully convincing that this mechanism is relevant for neutralization.
27. In general, the discussion is too focused on methodology aspects but lacks in making the case if and why the results would be relevant in vivo.
28. Line 937: no dashed lines in figure
29. Line 603 "the fixed sample was incubated with 1:500 dilution of b12 antibody for 1h at RT. " the actual concentration of the Ab needs to be stated.
30. Fig 1 D-E: The schematics are too small, in particular the red and blue labels are hard to distinguish
31. Figure 1C: The direction of rotation is incorrect
32. Figure 2: A The schematic representation does not help in understanding the photobleaching approach. The fluorophores are depicted too small and cannot be distinguished
33. Figure 2D: number of photobleaching events over what time frame (350s or 200s)?
34. Figure 5C,D: Legends are missing. In addition terming all three states "Conformation" is misleading
35. Figure 4,5 : It is unclear which graphs the headers describe
36. Figure 1C and Figure 6D: inconsistent color scheme for gp120 gp41
37. Figures 5C-D & 6A-C, It is unclear to which plot the titles belong

Reviewer #2 (Remarks to the Author):

The molecular mechanisms by which HIV Env mediates the membrane fusion stage of HIV-1 entry into host cells remain to be fully understood. This study applies state-of-the-art biophysical approaches (two-photon FRET-FLIM and single-molecule step photobleaching) to characterize dynamics of interactions between gp120 subunits of the same Env trimer and between gp120 subunits belonging to adjacent Env trimers. By comparing these interactions for mature vs. immature virions, for free virions vs. virions bound to the host cell and by analyzing the effects of several HIV-1-neutralizing antibodies to gp120, the authors relate the observed changes to the distinct types of conformational states of Env, including closed and open conformations of gp120 and Env clusters. As expected, application of soluble CD4 and virion binding to the cell membrane of MT-4 T cells destabilizes the closed conformation of Env. Interestingly, based on FRET-FLIM analysis, the authors conclude that HIV-1 maturation affects Env clustering but not intramolecular Env conformations. The experimental techniques developed in this work emerge as promising approaches allowing a fast detection of the dynamic transitions between different conformational states of Env, including assembly and disassembly of Env clusters. Altogether, the authors present novel, interesting, well controlled, generally convincing data to support their conclusions. This work will have significant impact on the fields of HIV-1 entry and membrane fusion in general. Concerns with the manuscript are discussed below.

Specific comments.

1) While the FRET-FLIM experiments are clearly described, the single-molecule step photobleaching approach remains unclear to me. I have found no explanation for the key assumption that clustered Envs have higher probability of synchronous bleaching leading to fewer steps of bleaching. Why does the timing of quenching of one fluorophore molecule depends on the proximity of another fluorophore molecule? The authors support this assumption by citing Ref. 27. Apparently, they mean the following statement in Ref 27: "As the number of fluorescent molecules increases in a complex, direct counting of steps become challenging. This is due to the increased possibility of synchronous photobleaching of multiple fluorophores, leading to the underestimation of the total steps." To me, what is meant here (in Ref. 27) is that the higher the TOTAL number of fluorophores present in the fluorescent spot in the image, the more difficult it is to resolve the individual steps of bleaching and to count the fluorophores

in this spot that in Ref. 27 corresponds to a single complex. In contrast, Carlon-Andres et al. apparently illuminate and bleach the entire ~100 nm virion and, thus, analyze bleaching events from the entire virion, and, thus the photobleaching traces apparently report the TOTAL number of fluorescent molecules in an individual virion rather than in a cluster of Envs. The authors should discuss and substantiate their assumption that clustered Envs bleach in fewer steps than the same number of Envs evenly distributed on viral envelope.

2) As reported in Ref. 3 and discussed by the authors, maturation increases lateral mobility of the Env. Can the high apparent FRET efficiency and decreased lifetimes characteristic for mature vs. immature virions be explained by higher mobility of Envs rather than by their clustering?

3) It appears that all measurements have been carried out at the room temperature. HIV-1 Env-mediated fusion is drastically inhibited at the room temperature. Can the intramolecular and intermolecular conformations and the conformational transitions of Env at the physiological temperature be considerably different from those characterized here at the room temperature?

4) I am intrigued by the authors' conclusion that "Env clustering ... is disrupted upon binding of Env to CD4". Can the authors suggest/discuss the possible interpretation for these findings? What are these findings suggest about the nature of the Env-Env interactions in the clusters? Note that there are many studies suggesting clustering of fusion proteins at the fusion conditions rather than disruption of the clusters (for instance, Gibbons et al., PMID: 14737160).

5) In the lines 223 and 272, the authors state that their findings "suggest/suggesting a functional implication". In both cases, I am not exactly clear on what is meant. For instance, in line 223, does clustering of Envs promote or inhibit fusion/infection? Both Env-CD4 interactions that disrupt clusters and virus maturation that forms clusters are known to be required for infection.

Minor comments:

1) Fig. 2B, Y-axes are shown as 'Intensity in photons' and it appears that Atto 594 fluorophore photobleaches after absorbing just a single photon. Most of the fluorophores undergo many absorption/emission cycles before bleaching (for instance, a single GFP molecule emits more than 10,000 photons before photodestruction). Is it known for Atto 594 to be so unstable or am I missing something here?

2) Line 27. "or to neutralizing antibodies", should it be "or by neutralizing antibodies"?

3) Line 46. Something seems to be missing in the sentence "The current hypothesis for Env intramolecular dynamics HIV-1 Env would undergo three states during this process:"

4) Line 159 a typo 'high' not 'hight'

Leonid Chernomordik

20th July 2021

We thank you and the two Reviewers for their careful and thoughtful evaluation of the manuscript and suggestions. We have made our best attempts to address each of the comments below, often supporting our answers with additional experiments. For example, we performed functional assays to validate the use of labelled HXB2 pseudo-typed HIV-1 viruses to study Env dynamics, as well as neutralization assays to relate our observations on Env dynamics using FRET-FLIM with the inhibitory mechanism of antibodies. We have also performed additional experiments using STED microscopy to relate this study with previous observations regarding Env cluster formation. If we feel it would be impossible to address a specific question experimentally, the technical hurdles encountered and efforts made have also been described.

Reviewers' comments:

Reviewer #1 (Remarks to the Author):

Carlon-Andres and colleagues combined in their study single molecule bleach and FRET-FLIM to characterize the dynamics of the HIV-1 Env trimer. The authors demonstrate a clustering effect of Env at the surface of viral particles and show that these clusters are disrupted by CD4 and neutralizing antibodies.

The topic of the study is clearly of high interest and I appreciate the use of alternative methods to monitor the structural dynamics of the Env trimer. However, in its current state the manuscript raises many questions that need to be solved. The validity of the experimental approach is not clear. Advantages and pitfalls of single molecule FRET must be laid out explicitly and the experimental set up verified.

The authors should also put more effort to make the interpretation of their results accessible to non-FRET experts. The result section is a heavy read even for the most dedicated reader. Figures are very detailed and sometimes confusing. The manuscript would benefit from figures that summarize and shifting of original data to the supplement. Figures should be redesigned to use colors that are easy to distinguish, have spacing between sub-panels and include accurate legends.

Specific points:

1. The authors fail to clearly define and distinguish what they term clustered, open and closed conformations. The term conformation is seemingly arbitrary used throughout the paper for both intra- and intermolecular interactions. Judging from text and figures one gets the impression as if clusters and open/closed conformations are mutually exclusive. However, this is clearly not the case and needs to be corrected throughout the manuscript and figures.

The authors need to use more distinct terms to clarify the differences between inter- and intramolecular dynamics. While it is fitting with the FRET data to describe this as a sequential process, the in vivo situation calls for a more clear distinction between closed/open Env and cluster association/dissociation.

We agree with the reviewer that in the previous version of the manuscript the distinction between open , closed conformation and cluster could be interpreted as mutually exclusive dynamics. We have now clarified in lines 229-233 and further discussed in lines 453-460, that our FRET-FLIM system can only differentiate closed and open Env conformations when no Env-Env interactions are taking place within the viral envelope. However, this does not imply that intramolecular dynamics are not occurring in the context of cluster formation. As discussed in lines 448-460, further experiments will be needed to understand trimer dynamics within the cluster.

2. The title should not refer to “structure” as the work solely focuses on dynamics measurements.

We now included the title “Structure dynamics of HIV-1 Env trimers on native virions engaged with living T cells”. It is common in the literature to infer from FRET experiments structural information. Clearly we have contributed to this end suggesting a structural model (Fig. 5).

3. Line 50: “that exposes otherwise hidden, more conserved epitopes, increasing susceptibility for antibody recognition “ This is not clear. More conserved than what? bnAb epitopes on closed trimers are also highly conserved.

We have now removed the expression “more conserved epitopes” from Line 50.

4. Line 52: “intramolecular structure”: should this rather be “intramolecular interactions”?

We have now removed the term “intramolecular structure” and written “intramolecular dynamics” following the reviewer’s advice.

5. The introduction is too brief especially considering the broad audience of Nature Communications. Much of the information on previous results and state of research included in the discussion, would have been helpful in the introduction. The introduction needs information on antibodies and their dependence on different conformations, the influence of the maturation process on Env and references to Cryo-ET/EM studies of closed and open Envs.

We agree with the reviewer and the introduction in the newest version of the manuscript is now enlarged (Lines 37-93). We have introduced information about the influence of HIV-1 maturation on Env processing and included literature relating the maturation state of HIV-1 particles with Env dynamics and functionality. In the new version of the manuscript we have also extended the introduction on previous studies on Env structure utilizing CryoEM and crystallography as well as reports on antibodies and how they influence or stabilise different Env conformations.

6. Why are different strains used in different experiments? The choice seems arbitrary.

We thank the reviewer for this comment. We agree that using different methods and strains is confusing, therefore we have chosen the HXB2 strain and performed new functional assays to validate their use in terms of infection and fusogenicity. In terms of methodology, we have also focused solely on FRET-FLIM now; as this technique fully described Env dynamics both in-vitro and when HIV-1 virions were engaged in T cells (Fig.3-4). Also, we have chosen HIV-1 Env (HXB2) as it is widely used in the field and thus enables comparison of results derived in different laboratories. Furthermore, HXB2 Env labelled with GFP_{OPT} in the V4 loop has been shown to be functional in [Nakane et al., 2015] (Lines 112-114) and confirmed by us in our new data (new Fig. 1E-G).

7. GFP is quite big and to some extent flexible (see also discussion line 427 etc). Since the FRET-signal depends on the distance, this flexibility may lower the resolution between conformational states that can be distinguished.

The flexibility of the linker is crucial to account for another important aspect of FRET (the relative angle between the two dipoles undergoing FRET). In fact, even if FRET is normally considered a molecular ruler and therefore depends on the distance, this interpretation is wrong. If the rotation of both the donor and the acceptor is not free [Lakovicz, J. (2010) Principles of Fluorescence Spectroscopy.] a particular dipole-dipole orientation will have a dramatic effect on the FRET efficiency. We agree that the GFP_{OPT} is ~ 32KDa and relatively big and might influence FRET signal, nevertheless, we and others have now carried out a number of functional assays as compared with the WT showing the functionality of this particular labelling strategy (Fig. 1E-G and Fig. S5). With this information and taking into consideration the importance of the relative orientation of the dipoles in FRET our approach is validated. We agree with the reviewer, however, that the dynamic range of FRET might be altered or different relative to other approaches published in the literature employing other labelling strategies (Mothes lab referenced papers). We discuss this in the newest version of the paper also highlighting the importance of the dipole-dipole orientation and other important considerations (Lines 385-395). Indeed, to our knowledge, we are the first taken into consideration HIV-1 Gag + particles exclusively to our FRET-FLIM analysis. On the top of that, we also take into consideration both immature and mature enriched HIV-1 populations when examining intra and intermolecular Env interactions.

8. The authors need to show that GFP insertion is not influencing Env conformation /opening by itself. The authors reference Nakane et al. who demonstrated no influence on expression level and fusogenicity. However, this needs to be shown for the current Envs. Also the exposure of CD4i and V3 epitopes may be influenced by GFP insertion. The exposure of these and other Ab epitopes must therefore be probed experimentally and stability of the GFP-Env compared with wild type.

We thank the reviewer for this comment. We have performed functional assays testing the ability of the HIV-1 viruses pseudo-typed with HXB2 labelled with GFP_{OPT} in the V4 loop to infect TZM-bl reporter cells compared to HIV-1 viruses bearing wildtype HXB2 Env (Fig. 1E-G) and observed a mild impact on the average infectivity (20% drop) of labelled viruses, which

was not statistically significant. We have also included new neutralization experiments using the bNAbs included in FRET-FLIM analyses as well as sCD4 to test the ability of these ligands to access different Env epitopes and neutralize infection in TZM-bl reporter cells (new Fig. S5). We have also compared the neutralization effect on HIV-1 viruses pseudo-typed with HXB2 WT or labelled V4-GFPopt, JRFL and NL4-3 Env, and observed a different effect of antibodies or sCD4 depending on the Env strain tested. However, we observed a similar neutralization profile in HXB2 WT vs V4-GFPopt labelled viruses, confirming that GFPopt insertion does not significantly impact Env dynamics and functionality. We have further clarified it in the manuscript by writing: "To resolve both, intra- and intermolecular interactions, the V4 loop of the gp120 was selected, as the side location of this residue facilitates FRET to occur between different Env trimers without altering Env functionality" (Lines 397-402).

9. The authors analyze their "mature virion prep" using a permeabilization strategy that allows to judge that 40% of the "mature prep" are actually mature virions. Considering that the preps are not fully mature, the preps should be rather referred to as "mature enriched". The corresponding figure S1B does not show any error bars, therefore it is unclear in how far this percentage varies between preps. Also, the percentage of maturation could be time- and temperature-dependent. Was this controlled for?

We agree with the reviewer that we should privilege the term "mature enriched" and we have done so in the newest version of the manuscript. We have performed more permeabilization assays for a total of three independently prepared viral samples (in absence or presence of the HIV-1 protease inhibitor Saquinavir, SQV) and have included the results in new Fig. S1B adding the corresponding error bars.

10. The authors compare mature vs immature virion preps are compared. It is unclear from these experiments if the "mature preps" always contain mature and immature virions.

We agree with the reviewer, and yes mature preps are referred in the manuscripts as viruses containing both mature and immature viruses. We consider the results section has more clarity now that we have employed the term "mature enriched viral samples" throughout the manuscript.

Judging from Figure 1 and Figure S2 these seems not to be the case, there seems to be no signal from immature virions. How can these be? The authors need to explain how they select solely mature virions from their prep that only contains 40% mature. As permeabilization appears not to have been done in parallel to the FRET experiments to quantify the % mature virions, the authors need to clarify how maturation stage was defined in the experiments and controlled for.

The multiparameter plots showing FRET-FLIM results where mature particles were considered also contain immature particles (Fig 2, S2 and S3). In fact, in these results one can see that the impact of the mature virions is apparent in terms of FRET efficiency and lifetime as judged by the differences observed in mature-enriched viruses compared to only immature (SQV treated) in presence of donor and acceptor (Fig 1C compared to 1E). We have performed additional replicates (from a total of three independent preparations, Fig S1B) to better define the

maturation efficiency of “mature-enriched” samples and describe it in the manuscript in lines 124-136.

11. Why is the FRET-background assessed only with the donor and not with the acceptor (Figure S2)?

In the 2-dimensional graphs the donor average lifetime $\langle\tau_D\rangle$ is plotted against the apparent FRET efficiency (E_{app}) calculated with the sensitized emission procedure as specified in the methods section. The no-FRET situation has been determined for the mature-enriched and immature viral samples in presence of the donor alone to ensure no FRET is occurring between fluorophores. Hence, a no/low-FRET situation in presence of the FRET couple could be related to a specific Env state.

Should not capture of the FRET-signal be done at the acceptor wavelength (longer than the donor)?

In the case of FLIM (y axis) the donor lifetime is enough to calculate the true FRET efficiency and the fraction of interacting donor independently [Padilla-Parra et al., 2008 Biophys J].

In the case of the apparent FRET efficiency calculated with intensity images (x axis) we did collect both the donor emission (at 500-500nm) and the acceptor sensitized emission (600 – 650nm). Background corrected images were produced from both stacks of images and then kernel 2-dimensional probability maps produced with FRET images compared with lifetime images.

It was important to compare two different ways of detecting FRET (lifetime and intensity-based sensitized emission) to be certain that both lifetime shortening and increase of E_{app} were correlated. This powerful approach offers a built-in control that further re-enforces our approach as other environmental effects on the lifetime due to the orientation of the dipoles or changes in diffraction that may cause lifetime changes not strictly related to FRET will also be taken into consideration but will not influence our FRET interpretation.

It needs to be shown whether or not GFPopt in Env causes background FRET by transmitting to the acceptor.

Indeed, we did not produce the condition for HIV-1-HXB2-GFPopt-NBAtto594 (Excitation 950nm) while measuring the donor lifetime and the E_{app} . We estimated that this condition was not needed for two reasons:

- 1- We are comparing mature and immature virions with NB488 (donor) and NB488 + NB594 (donor + acceptor). In both cases our negative controls are labelled exactly the same way (Fig. S2 A -B). The difference being only the addition of the acceptor nanobody (NB594). We reasoned that the relative difference obtained in FRET was therefore strictly related to the presence of the acceptor.*
- 2- When performing the saponin assay (Fig. S1) one can see that the addition of saponin in mature viruses eliminates the contribution of Gag-GFP and the only signal that remains in these viruses is the NB594. Indeed, with the very same conditions our HyD*

detectors could not detect photons from GFPopt after the saponin assay (i.e. Gag-GFP release) indicating that in these cases the GFPopt is merely employed as a scaffold for the nanobodies.

12. In the discussion (lines 423-425) the authors state that mature and immature particles can be discriminated by GFP-lifetime- How was this assessed? It is generally unclear why a difference in the GFP lifetime between mature and immature particles would exist. The authors explain that the lifetime of the signal is reduced in Fig 1D due to the FRET. In Fig. 1E (left) they also distinguish some FRET signal in the intermediate range, but here the lifetime appears longer (compared to Fig S2B, right panel)?

We have found that measuring the average lifetime for mature-enriched and immature viral samples that are labelled exactly in the same way, one can ascertain mature from immature particles by checking Gag-GFP lifetimes. We have performed further experiments (n=3) relating the lifetime of GFP-Gag from individual viral particles before adding saponin and the release of GFP content 10 min after saponin addition. Hence, we have classified particles as mature or immature depending on their ability to release the internal GFP as a result of Gag processing during maturation. As shown in the figure below, mature particles from the mature-enriched viral sample had a distinct average lifetime compared to immature particles from the same viral sample or compared to immature particles prepared in presence of SQV. These measurements show that mature and immature viruses present different spectral signatures where higher average lifetimes are found for mature particles.

We have however decided not to include these results in the current manuscript as the signal from Gag-GFP was not employed in the assays to filter out mature from immature particles in the mature-enriched sample. Instead, relative changes in lifetime and apparent FRET efficiencies in presence of donor alone or with the FRET-pair fluorophores were used to assess the Env dynamic landscape.

We consider that these results are very interesting to improve future analysis of Env dynamics in mature vs immature particles. Differences in Gag-GFP lifetime in mature vs immature particles although apparent in terms of population, are not clear enough at a single particle level. In fact, an improved construct using Gag or Vpr HIV-1 proteins as a biosensor for viral maturation would refine the analysis of Env dynamics as a function of maturation.

Fig 1 Rebuttal. Lifetime measurements of GFP-Gag in mature-enriched and immature HIV-1 preparations.

13. The FRET results with Envs containing GFPopt in V1 stick out: Here the donor and acceptor should in theory be located more closely together (at least in the closed state) compared to labelling on V4. This should lead to a stronger FRET compared to the labelling on V4. But this is not seen in the data. The authors need to explain this.

The reviewer might be right with respect the intramolecular Env interactions, and we detect intra-molecular FRET for both mature and immature particles for Envs GFPopt in V1 (Fig. S3). Interestingly, these results, serve as a robust control to verify that high FRET values come from intermolecular FRET as in this case $E_{app} > 0.23$ is lost (Fig. S3B compared to Fig.2C). Indeed, labelling of the V1 loop instead of V4 would make the distance between Envs larger than 10nm even when Env is organized in clusters (mature virions). We agree with the reviewer that introducing the FRET pair in the apex of Env would eventually decrease the intramolecular distance between donor and acceptor fluorophores thereby facilitating FRET to occur. However, concerning the efficiency of FRET, we hypothesized that higher efficiencies could not be explained solely by intramolecular FRET but also and more importantly by the presence of higher number of acceptors/donor [Godet et al. 2019], as is the case when several Envs are in close proximity. Even if clustering could occur in spite of the V1 labelling, intermolecular distance between labels on Env trimers is increased and therefore high FRET efficiencies are less likely to be reached. We have explained it in the revised manuscript by writing: "This labelling approach that positions the donor and acceptor fluorophores proximal to the apex of Env when adopting a closed conformation, is expected to increase the distance between different Env trimers, thereby minimizing the number of acceptor molecules per donor which would drastically reduce or eliminate the intermolecular FRET", in lines 208-212.

14. In the experiments with b12 the authors first incubate the viral particles with sCD4, then fix, and add the labelled b12. As the authors state themselves (lines 341-344), b12 is unable to bind the closed conformation and itself stabilizes an intermediate/open state. Since the cells are fixed, b12 should not be able to induce opening and residual binding should be weak and suffer from an increased off-rate which may lead to an overestimation of photobleaching events. These aspects should be considered in the interpretation of the results. The authors further state that they use a saturating sCD4 concentration but still use the CD4bs Ab b12 as a secondary stain. Since substantial competition should occur, the authors need to explain the rationale of this setup and how to interpret measurements more clearly.

We thank the reviewer for this comment. We have decided to exclude experiments using single-molecule photobleaching from the current manuscript. Please, see further explanation about this decision in the reply to comment #1 of Reviewer2.

15. In the experiments with bnAbs (Figure 3 and 4) the label for photobleaching is attached to the bnAbs. How is the labeling and photobleaching done for the reference in the absence of antibody (since JR-FL and NL4-3 do not contain GFPopt the labelled nanobody cannot be used instead)?

As stated in previous comment #14, we have decided to exclude experiments using single-molecule photobleaching. We have explained in detail this decision in reply to comment #1 of Reviewer2.

16. The authors conclude that Env cluster formation requires a closed conformation. However, even in the presence of PGT145 cluster formation is apparently reduced. This should be discussed.

We have observed that transitions towards Env-Env interactions occurred via Env closed conformation. We do not know however if this is a requirement for cluster formation or whether an open conformation can be adopted in the context of clusters. As discussed in lines 453-460, further studies would be required to answer these questions. The observation of Env-Env interaction disruption upon PGT145 however, could be an effect unrelated to the specific intramolecular conformation adopted by the trimer i.e. be a consequence of PGT145 sterically impairing inter-FRET upon Env binding.

17. In Figure 1D and Figures 5 and 6, the authors look at the FRET signal below 0.12 (attributed to the open state). However they defined in Figure S2A the threshold for background at 0.1. How is the signal below 0.12 accurately quantified?

Results in Fig. S2 correspond to viral samples labelled with the donor alone. In absence of acceptor FRET cannot occur. The conditions in Fig. 1C-F, 3 and 4 are all in presence of the FRET-pair. In these conditions, FRET can occur between fluorophores and in fact, we observed three different FRET regimes referred as: no/low, intermediate and high. Our hypothesis is that no/low FRET regimes in this context is related to a situation in which FRET could potentially occur between both fluorophores but we reasoned that absence of FRET in this case relates to a specific Env conformation. This hypothesis was confirmed upon binding of sCD4, in which we observed an enrichment of the no/low-FRET situation (Fig. 2D and F). Therefore we related FRET regimes below 0.12 to the open Env conformation.

18. Line 125: A saturating concentration of sCD4 leads to an increase of the no-FRET status. Is this also influenced by shedding of Env? At saturating sCD4 doses this should happen to some degree? At 10ug/ml CD4 shedding of HxB2 Env would be considered highly likely (Figures 1 and 2).

We thank the reviewer for this comment. Although shedding of Env after sCD4 binding can happen to a certain degree, it is unlikely our FRET-FLIM results reflect this phenomenon. Our rationale behind is that only viral particles showing a positive signal in both, 500-550 nm (donor) and 600-650 nm (acceptor) channels have been selected for further analyses. Co-localisation in both channels implies that the V4 loop label in gp120 has been detected by nanobodies targeted against GFPopt and therefore, the localization of Env trimers within the viral envelope is preserved.

Minor comments:

19. Line 260: It would be important to bring the effects of cluster disruption into context with the binding/neutralization capacity of the respective antibodies and the used isolates. Antibody concentration used need to be stated.

In the new version of the manuscript we have not included the results of photobleaching experiments, as on one hand the new experiments performed with a TIRF system the statistical significance is not as good as the one presented previously. On the other hand, we consider that the photophysical aspects of single-molecule photobleaching in this context need further characterization. We have now stated antibody concentrations in multiple places in the revised manuscript, for example, in the case of experiments with T cells (Fig. 4) we have specified the concentration of antibodies used (Line 316) as well as in the material and methods section. In the case of the new functional assays (New Fig. S5), different antibody concentrations have been indicated in the graph.

20. Line 269: It is confusing that both cluster formation and intramolecular dynamics are termed “conformation”. Change to cluster formation?

We thank the reviewer for this comment. We agree that the term conformation referred to intermolecular dynamics is confusing. We have favoured the concept intermolecular interactions instead of cluster formation, as the resolution of our FRET-FLIM assay does not allow to determine the distribution of Envs within the viral membrane, but does reflect the interaction between different Env molecules. To better define and illustrate this concept we have used STED super-resolution microscopy following the method developed by [Chojnacki et al., 2012]. We have been able to define different Env pattern distributions for immature and mature enriched badges using STED microscopy (new Fig. S4) and classified them as in the original paper as “Cluster”, “Intermediate” or “Sparse”.

21. Line 320: This sentence is unclear

We have rephrased the sentence “We have thus shown that HIV-1 Env transitions towards an open conformation with longer CDF half lifetimes when engaged to T cells as compared to in vitro virions” and now have written “ We have thus shown that HIV-1 Env open conformation is more stable in virions engaged to T cells compared to cell-free virions”, in new Line 281-282.

22. Line 319: potential shedding influence needs to be discussed

We have discussed this aspect in response to comment #18

23. Line 327: the differentiation into open, closed and clustered as separate conformations is unclear. Why can Env in open conformation not form clusters?

We agree with the reviewer the differentiation into open, closed and clustered state of Env is confusing. We have now modified the text accordingly in lines 288-292.

24. Line 346: “PGT145 exhibits a tendency to destabilize the open Env”. The authors state this

to explain their FRET finding. However this statement implicates that PGT145 binds and disturbs open Env. Since PGT145 binds closed Env a predominant action on open Envs is not likely. This needs to be clarified.

We agree with the reviewer that this sentence is confusing. It has been now corrected in line 317-319.

25. The two result sections describing bnAb effects should be merged

In the new version of the manuscript, we have included the results describing the effects of bNAbs on viral infectivity (Fig. S5) and Env dynamics (Fig. 4) in the same section from Line 294 to 341.

26. The discussion lacks an interpretation on the effect of bnAbs on cluster formation: It is not fully convincing that this mechanism is relevant for neutralization.

We thank the reviewer for this comment. We agree it is of importance to relate the effect of bNAbs on Env dynamics to its neutralization ability. We have performed neutralizing experiments and the results have been included in Fig. S5. We have observed different neutralization abilities of the three antibodies tested depending on the HIV-1 strain. In case of HXB2 pseudotyped viruses (irrespective of bearing the GFP_{OPT} label at the V4 loop or not) antibodies known to stabilize an open/intermediate conformation (b12 and 10E8) and disrupting intermolecular dynamics induced a strong impairment of HIV-1 infectivity. This was however not the case for the PGT145, which has a preferential binding to the closed Env conformation. We have now included a more thorough discussion about these results in lines 461-477.

27. In general, the discussion is too focused on methodology aspects but lacks in making the case if and why the results would be relevant in vivo.

We have now included a more detailed discussion relating aspects of the current results observed using our FRET-FLIM system and the situation in vivo. For example, we have now included a discussion about the limitation of our system to observe intramolecular conformations in the context of Env-Env interactions (lines 454-460). We have also related the effect of neutralizing antibodies and CD4 binding on viral infectivity with the Env intramolecular (lines 461-478) and intermolecular dynamics (lines 479-493) observed in FRET-FLIM analyses.

28. Line 937: no dashed lines in figure

White dashed lines in Fig. 3A (left panel) indicate the region of the image magnified in right panels.

29. Line 603 “the fixed sample was incubated with 1:500 dilution of b12 antibody for 1h at RT. “ the actual concentration of the Ab needs to be stated.

Concentrations in $\mu\text{g}/\text{mL}$ employed in FRET-FLIM analyses and neutralization assays have now been included in the results and material and methods sections.

30. Fig 1 D-E: The schematics are too small, in particular the red and blue labels are hard to distinguish

We have now enlarged the schematics shown next to each kernel graph of FRET-FLIM analyses in Fig. 2, S2 and S3.

31. Figure 1C: The direction of rotation is incorrect.

We have now corrected the direction of rotation.

32. Figure 2: A The schematic representation does not help in understanding the photobleaching approach. The fluorophores are depicted too small and cannot be distinguished

33. Figure 2D: number of photobleaching events over what time frame (350s or 200s)?

Results relative to photobleaching experiments have not been included in the new version of the manuscript as reasoned in response to Reviewer2.

34. Figure 5C,D: Legends are missing. In addition terming all three states "Conformation" is misleading. 35. Figure 4,5 : It is unclear which graphs the headers describe 37. Figures 5C-D & 6A-C, It is unclear to which plot the titles belong

We have now included the legends in a box above the graphs in new Fig. 3 and 4 and included headers in each graph of new Fig. 4. We have replaced the term "Cluster conformation" with "Env-Env interactions" in Fig. 3,4 and 5.

36. Figure 1C and Figure 6D: inconsistent color scheme for gp120 gp41

We have now used homogeneous colour scheme throughout the figures (Fig. 1, 4 and 5).

Reviewer #2 (Remarks to the Author):

The molecular mechanisms by which HIV Env mediates the membrane fusion stage of HIV-1 entry into host cells remain to be fully understood. This study applies state-of-the-art biophysical approaches (two-photon FRET-FLIM and single-molecule step photobleaching) to characterize dynamics of interactions between gp120 subunits of the same Env trimer and between gp120 subunits belonging to adjacent Env trimers. By comparing these interactions for mature vs. immature virions, for free virions vs. virions bound to the host cell and by analyzing the effects of several HIV-1-neutralizing antibodies to gp120, the authors relate the observed changes to the distinct types of conformational states of Env, including closed and open conformations of gp120 and Env clusters. As expected, application of soluble CD4 and virion binding to the cell membrane of MT-4 T cells destabilizes the closed conformation of

Env. Interestingly, based on FRET-FLIM analysis, the authors conclude that HIV-1 maturation affects Env clustering but not intramolecular Env conformations. The experimental techniques developed in this work emerge as promising approaches allowing a fast detection of the dynamic transitions between different conformational states of Env, including assembly and disassembly of Env clusters. Altogether, the authors present novel, interesting, well controlled, generally convincing data to support their conclusions. This work will have significant impact on the fields of HIV-1 entry and membrane fusion in general. Concerns with the manuscript are discussed below.

Specific comments.

1) While the FRET-FLIM experiments are clearly described, the single-molecule step photobleaching approach remains unclear to me. I have found no explanation for the key assumption that clustered Envs have higher probability of synchronous bleaching leading to fewer steps of bleaching. Why does the timing of quenching of one fluorophore molecule depends on the proximity of another fluorophore molecule? The authors support this assumption by citing Ref. 27. Apparently, they mean the following statement in Ref 27: "As the number of fluorescent molecules increases in a complex, direct counting of steps become challenging. This is due to the increased possibility of synchronous photobleaching of multiple fluorophores, leading to the underestimation of the total steps." To me, what is meant here (in Ref. 27) is that the higher the TOTAL number of fluorophores present in the fluorescent spot in the image, the more difficult it is to resolve the individual steps of bleaching and to count the fluorophores in this spot that in Ref. 27 corresponds to a single complex. In contrast, Carlon-Andres et al. apparently illuminate and bleach the entire ~100 nm virion and, thus, analyze bleaching events from the entire virion, and, thus the photobleaching traces apparently report the TOTAL number of fluorescent molecules in an individual virion rather than in a cluster of Envs. The authors should discuss and substantiate their assumption that clustered Envs bleach in fewer steps than the same number of Envs evenly distributed on viral envelope.

We thank the reviewer for this important comment. We have tried to address this criticism by performing more experiments on single molecule photobleaching, this time employing a TIRF approach and a very sensitive back-illuminated camera adapted for single molecule detection (prime 95B). We have taken different HIV-1 (Gag-GFP) virions pseudotyped with different Env strains (HXB2, JRFL and NL4.3) immunolabelled using b12 as primary antibody and secondary antibodies coupled to Atto594. Fast double color TIRF imaging with mild 488 and high power 594 nm laser induced single particle photobleaching only for the red channel and individual traces were analyzed manually only for particles with Gag-GFP+ signal. We have analyzed both mature and mature enriched HIV-1 samples and found a similar pattern as the one presented in the first version of the manuscript:

Fig 2 Rebuttal. Number of single Env molecules for ($n = 30$) particles per condition utilizing a single molecule photobleaching TIRF approach.

Even if these new data corroborates our previous observations, the statistical significance is not as good as the one presented previously. This fact, and Reviewer's 1 comment made us think that perhaps including these experiments the article was too confusing. Therefore, we have decided to present only the FRET-FLIM data and focus on one single Env strain (HXB2) and one single approach FRET-FLIM. We acknowledge that further research needs to be performed to fully define the photophysical aspects of this bleaching, even if when comparing mature vs immature particles the differences are consistent across different strains and labelling strategies and this approach is able to unveil Env clusters.

2) As reported in Ref. 3 and discussed by the authors, maturation increases lateral mobility of the Env. Can the high apparent FRET efficiency and decreased lifetimes characteristic for mature vs. immature virions be explained by higher mobility of Envs rather than by their clustering?

Indeed $2\text{nm}^2/\text{sec}$ [Chojnacki et al. 2017] could have an impact in the increased FRET efficiency observed in mature particles. We have presented, however, the time-traces for individual HIV-1 particles when engaged in live T cells and one can measure the kinetics of Env clustering – association and dissociation (Fig. 3B). The simplest explanation for changes in high FRET efficiency (Intermolecular FRET) and how it transitions to low FRET (open Env conformation) or moderate FRET (closed Env) is that Env is clustered. Another explanation could be sudden increase in Env, nevertheless considering isotropic biophysical Env behaviour in mature particles, this possibility is less likely.

3) It appears that all measurements have been carried out at the room temperature. HIV-1 Env-mediated fusion is drastically inhibited at the room temperature. Can the intramolecular and intermolecular conformations and the conformational transitions of Env at the

physiological temperature be considerably different from those characterized here at the room temperature?

Yes, the HIV-1 fusion reaction is temperature dependent as shown in (Henderson and Hope, 2006). It has previously been shown, however, that receptor priming can occur at 4 °C and that temperature-arrested state (TAS) can generate a stable intermediate that is kinetically predisposed to complete the fusion event leading to infection (Henderson and Hope, 2006). In our case all experiments were carried out at room temperature and therefore receptor priming and the first steps of the fusion reaction can proceed. It would be very interesting to relate these findings with productive HIV-1 fusion at 37 °C. We note, that other biophysical studies on single HIV-1 particles were always carried out at room temperature [Munro et al. 2014, Chojnacki et al. 2017]. In all cases, we agree with the reviewer that this is a crucial point that we want to address in future studies.

4) I am intrigued by the authors' conclusion that "Env clustering ... is disrupted upon binding of Env to CD4". Can the authors suggest/discuss the possible interpretation for these findings? What are these findings suggest about the nature of the Env-Env interactions in the clusters? Note that there are many studies suggesting clustering of fusion proteins at the fusion conditions rather than disruption of the clusters (for instance, Gibbons et al., PMID: 14737160).

5) In the lines 223 and 272, the authors state that their findings "suggest/suggesting a functional implication". In both cases, I am not exactly clear on what is meant. For instance, in line 223, does clustering of Envs promote or inhibit fusion/infection? Both Env-CD4 interactions that disrupt clusters and virus maturation that forms clusters are known to be required for infection.

We thank the reviewer for comments #4 and #5. We have included a paragraph (Lines 405-424 and 480-493) in the newest version of the manuscript discussing our views on the importance of Env-Env interactions during HIV-1 priming (pre-fusion reaction) that further illustrates the model depicted in Fig. 5. In brief, our interpretation is that the cluster is needed for priming, but upon Env-CD4 interaction this cluster is disrupted (at least based on FRET resolution) more than 10 nm apart. This vision is further supported by our previous paper (Iliopoulou et al., 2018 NSMB) and also a recent paper from the Bjorkman lab (2020, eLife) where 1 or two spokes were seen to be enough to start the fusion reaction.

Minor comments:

1) Fig. 2B, Y-axes are shown as 'Intensity in photons' and it appears that Atto 594 fluorophore photobleaches after absorbing just a single photon. Most of the fluorophores undergo many absorption/emission cycles before bleaching (for instance, a single GFP molecule emits more than 10,000 photons before photodestruction). Is it known for Atto 594 to be so unstable or am I missing something here?

As explained in response to comment #1 we have not included results from single-molecule photobleaching experiments in the new version of the manuscript.

2) Line 27. "or to neutralizing antibodies", should it be "or by neutralizing antibodies"?

This has now been corrected in Line 27.

3) Line 46. Something seems to be missing in the sentence "The current hypothesis for Env intramolecular dynamics HIV-1 Env would undergo three states during this process:"

This sentence has been removed in the new version of the manuscript.

4) Line 159 a typo 'high' not 'hight'

This has now been corrected.

Leonid Chernomordik

Thank you, Dr. Chernomordik!

REVIEWERS' COMMENTS:

Reviewer #1 (Remarks to the Author):

The authors have addressed all main concerns.

Reviewer #2 (Remarks to the Author):

This study develops novel approaches for analysis of the dynamic transitions between different conformational states of HIV Env and shows that lateral Env-Env interactions in protein clusters are disrupted upon binding of Env to CD4. This work will have significant impact on the fields of HIV-1 entry and membrane fusion in general. The revision has strengthened this interesting and important study and addressed almost all my comments/questions. I have only one remaining suggestion related to my 3rd comment of the 1st round. The analysis of the conformations of Env leading to the conclusion that Env-CD4 engagement induces dissociation of Env clusters is carried out at the room temperature and, thus, under conditions not-permissive for Env-mediated fusion. The authors note that "other biophysical studies on single HIV-1 particles were always carried out at room temperature". This is an important consideration. However, I think the readers will benefit from a short comment in the Discussion addressing the applicability of these findings to intramolecular and intermolecular conformational transitions of Env at the physiological fusion-permissive temperature.